# Self-evolving LLM agents with in-distribution Optimization

Yudi Zhang [1]   Meng Fang [2 1]   Zhenfang Chen [3]   Mykola Pechenizkiy [1]

## Abstract

Large Language Models (LLMs) have recently emerged as powerful controllers for interactive agents in complex environments, yet training them to perform reliable long-horizon decision making remains a fundamental challenge. A key difficulty lies in credit assignment: agents often receive delayed rewards only at the end of episodes. In this paper, we propose Q-Evolve, a self-evolving framework for LLM agents that unifies automatic process-reward labeling and policy learning within a principled in-distribution reinforcement learning paradigm. In each evolving iteration, our method learns an in-distribution critic from a hybrid off-policy dataset that combines expert demonstrations with agent-generated trajectories, stabilizing Bellman backups in sparse-reward settings via a weighted Implicit Q-Learning objective. The learned value function is then used to derive step-wise process rewards through advantage estimation, enabling dense and reliable supervision without environment backtracking or human annotation. Leveraging these signals, we perform behavior-proximal policy optimization that evolves the agent over the data used for process reward labeling, allowing iterative self-improvement without exacerbating distribution shift. We evaluate our method on AlfWorld, WebShop, and ScienceWorld, showing Q-Evolve outperforms strong baselines in sample efficiency, robustness, and overall task performance. Our results demonstrate that stable agent self-evolution is achievable through the co-evolution of process-level supervision and policy, both grounded within a shared in-distribution learning loop. Project Page.

---

[1]Eindhoven University of Technology, Netherlands [2]University of Liverpool, United Kingdom [3]MIT-IBM Watson AI Lab. Correspondence to: Meng Fang <Meng.Fang@liverpool.ac.uk>.

*Proceedings of the 43$^{rd}$ International Conference on Machine Learning*, Seoul, South Korea. PMLR 306, 2026. Copyright 2026 by the author(s).

## 1. Introduction

While Large Language Models (LLMs) (Achiam et al., 2023; Zhao et al.), have demonstrated exceptional reasoning capabilities (Wei et al., 2022; Yao et al., 2023), their role is increasingly shifting from static text generation to driving interactive agents (Wang et al., 2024a). These agents must move beyond simple prediction to master sequential decision-making in dynamic environments. By leveraging the cognitive power of LLMs, researchers have explored their potential in various interactive domains, including navigation (Song et al., 2024; Lin et al., 2025; Feng et al., 2025), gaming (Wang et al.; Liu et al.), and robotics (Kim et al.; Wang et al., 2025). However, achieving consistent and closed-loop autonomy remains a significant challenge, as LLMs must effectively bridge the gap between high-level reasoning and reliable execution.

A central challenge in training LLM agents for interactive, long-horizon tasks is that feedback is often sparse and severely delayed (Lin et al., 2025; Feng et al., 2025). Agents typically receive meaningful supervision only at episode termination, making it difficult to attribute success or failure to individual intermediate decisions (Arjona-Medina et al., 2019; Ren et al., 2022; Zhang et al., 2023a). To address this, recent efforts aim to automatically derive step-wise process rewards to avoid expensive, hard-to-scale manual annotation (Lightman et al., 2023; Ma et al., 2023). For example, Wang et al. (2024b;c); Lin et al. (2025) rely on extensive online interaction to search and estimate Q-values as process reward labels, while GiGPO (Feng et al., 2025) introduces anchor-state grouping to calculate step-level advantages.

A more fundamental limitation shared by existing approaches lies in the reliability of the feedback they consume. In particular, process-level supervision is inherently *distribution-sensitive*: process rewards are only reliable on the similar state-action distribution on which they are derived. However, as illustrated in Figure 1, during online policy optimization or test-time scaling, the evolving policy inevitably generates unseen actions even when the starting state is sampled from the PRM's training set. This issue is further exacerbated in dynamic environments. As the agent interacts with the environment, the underlying environmental dynamics propel the agent into out-of-distribution states, might leading to a catastrophic distribution shift that in-

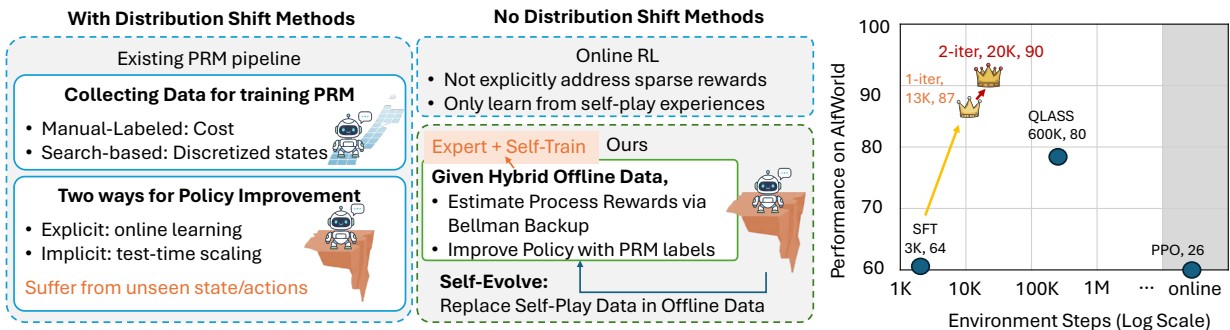

*Figure 1.* Comparison of existing methods. **Left:** Existing PRM methods rely on costly manual labels or search-based rollouts requiring discrete states, often failing due to distribution shifts between PRM training and policy improvement. **Upper Mid:** Most online RL does not address episodic sparse rewards. **Bottom Mid:** Our framework utilizes a hybrid off-policy dataset (expert + agents' interaction data) to derive rewards via Bellman backups. By co-evolving process reward supervision and policy improvement within a shared in-distribution loop, the agent achieves stable self-evolution. **Right:** A visualization of performance *vs* environment steps required for collecting data.

validates the PRM's feedback. Additionally, most existing frameworks rely on restrictive assumptions, such as environment determinism, the availability of state-backtracking, or discretizable state features for grouping and searching. These assumptions, coupled with the requirement for exhaustive online interactions, significantly hinder the deployment of such methods in realistic, high-stakes, or non-deterministic scenarios. Therefore, there is a growing need for methods that both generate and leverage step-wise supervision within the same distribution to keep the process reward labeling reliable.

A natural way is to use classical Bellman backups in an offline manner, which theoretically addresses long-horizon credit assignment. However, transferring this paradigm to LLM agents remains difficult. First, in the general episodic reward settings, the bootstrapping mechanism is prone to significant stochastic noise that accumulates without intermediate signals, preventing stable convergence. This is further compounded by the combinatorial action space of LLMs, where scalar Q-values defined over multi-token sequences fail to facilitate direct policy optimization. Consequently, rather than optimizing the policy directly, existing offline RL works for LLMs often resort to using external critics learned from offline data to re-evaluate or calibrate candidate actions (Snell et al., 2023; Xiang et al., 2024). While helpful, these approaches treat the critic as an auxiliary filter rather than an intrinsic objective. This reliance on external calibration has two drawbacks: it prevents the LLM from becoming a self-contained agent that can be further evolved, and it leaves unaddressed the distribution shift between the offline training data and the policy's own distribution at test time.

To address these limitations, we propose Q-Evolve, a self-evolving framework for training LLM agents that unifies automatic process-reward acquisition and in-distribution policy optimization within a closed learning loop. Unlike

*Table 1.* Comparison with the alternative methods. ✓ = explicitly supported. ✗ = not supported. SE: self-evolve; N-DS: no need for discrete state; Bellman: Bellman backup for credit assignment; PR: assign process rewards. N-EO: no need for extensive online interaction. Self-train: learn from self-collected experiences.

| Method | SE | N-DS | Bellman | PR | N-EO | Self-Train |
|---|---|---|---|---|---|---|
| BC | ✗ | ✓ | ✗ | ✗ | ✓ | ✗ |
| PPO | ✗ | ✓ | ✓ | ✗ | ✗ | ✓ |
| ETO | ✗ | ✓ | ✗ | ✗ | ✓ | ✓ |
| QLASS | ✗ | ✗ | ✓ | ✓ | ✗ | ✓ |
| GiGPO | ✗ | ✗ | ✗ | ✓ | ✗ | ✓ |
| Q-Evolve | ✓ | ✓ | ✓ | ✓ | ✓ | ✓ |

prior pipelines that rely on static reward models or one-shot offline critics, our framework enables the agent to iteratively improve itself by deriving step-wise supervision from an evolving Q-based critic and updating the policy via behavior-proximal policy optimization to avoid distribution shift. Crucially, this design allows policy, critic, and data to co-evolve, while each policy update remains grounded within a hybrid offline dataset, thereby mitigating distribution shift and stabilizing long-horizon credit assignment. We compare our method with alternatives in Table 1. To address sparse and delayed feedback, we derive step-wise supervision by learning an in-distribution value function on a hybrid dataset that combines expert demonstrations with agent-generated trajectories. This mixture is crucial for stability: Bellman backups on purely self-generated data can be dominated by stochastic noise, especially when success rates are low, whereas expert data provides sparse-signal grounding that stabilizes the value targets. Specifically, we estimate an in-distribution critic (Kostrikov et al., 2022), utilizing a weighted Implicit Q-Learning objective to address episodic rewards. Rather than training a standalone PRM, we derive advantage estimation as process rewards via Generalized Advantage Estimation (GAE) (Schulman

et al., 2016) for policy learning. This approach utilizes Bellman propagation to "fill in" missing intermediate rewards without requiring environment backtracking or external human labeling. With the step-wise process reward signals obtained, the policy is then updated via a behavior-proximal policy optimization (Zhuang et al., 2023), aiming to amplify beneficial actions and suppress harmful ones. To further promote generalization, we introduce a more permissive clipping on the objective's lower bound, allowing for larger policy updates when suppressing bad responses.

In summary, our work makes the following contributions. First, we propose Q-Evolve, a self-evolving framework that jointly performs automatic process reward labeling and in-distribution policy learning, keeping the agent's single-step policy improvement strictly within fixed hybrid off-policy data while enabling iterative improvement toward optimal long-horizon behavior via critic updates. Second, to handle extremely sparse and delayed rewards, we train an in-distribution critic with a weighted Implicit Q-Learning objective over a hybrid dataset, which mixes expert demonstrations with on-train agent trajectories. Third, we perform in-distribution policy learning via behavior-proximal policy optimization, along with down-weighting thinking and clipping lower to amplify positive-advantage tokens and explicitly suppress negative-advantage ones. Finally, we evaluate Q-Evolve on AlfWorld, WebShop, and ScienceWorld, achieving consistent improvements over strong baselines in sample efficiency and effectiveness.

## 2. Related Work

In this section, we review the LLM agents, process reward models, and self-evolving agents.

**LLM agents.** Driven by the rapid advancement of LLMs, language-based agents have demonstrated strong performance across diverse domains, such as code (Roziere et al., 2023), math (Luo et al., 2023; Yuan et al., 2023), game (Wang et al.; Liu et al.; Fang et al., 2024; Zhang et al., 2025), computer use (Hong et al., 2024; Liu et al., 2026; Wang et al., 2025) and robotics (Wang et al., 2025). By using natural language both to reason and to interact with environments, these agents can generalize across tasks and provide greater flexibility than traditional reinforcement learning agents (Yao et al., 2022b; Shinn et al., 2023). Recent work further extends their capabilities through planning (Song et al., 2023; Zhao et al., 2023; Chen et al., 2025), and tool use (Yuan et al., 2025; Lu et al., 2025), expanding the range of settings where they can be applied. Nevertheless, achieving reliable long-horizon decision-making remains difficult, with persistent issues such as sparse rewards, credit assignment, and poor sample efficiency (Arjona-Medina et al., 2019; Ren et al., 2022; Zhang et al., 2023a; Feng et al., 2025).

**Process reward models** (PRMs) have been studied mainly for multi-step reasoning problems, such as mathematical problem solving (Cobbe et al., 2021), where they provide step-level supervision on intermediate reasoning instead of relying only on final outcome rewards (Lightman et al., 2023; Uesato et al., 2022). Early PRMs were trained with human-annotated process supervision (Uesato et al., 2022), while more recent work explores computing PRMs automatically, e.g., by treating them as Q-value estimates (Wang et al., 2024b; Luo et al., 2024). PRMs have been used both to train generators (Shao et al., 2024) and to enable test-time scaling via beam search (Snell et al., 2024), heuristic search (Ma et al., 2023), or tree search (Wu et al., 2024). In contrast, PRMs are much less discussed in LLM agent settings (Lin et al., 2025; Choudhury, 2025). Lin et al. (2025) extract Q-value labels by constructing an exploration tree and Bellman backup as the process reward modeling, while Choudhury (2025) builds AgentPRM and InversePRM within an RLHF framework and highlights key challenges and opportunities for PRMs in LLM agents. However, those methods all overlook the risk of the distribution shift.

**Self-evolving agents.** Recently, self-evolving agents have gained significant attention, shifting the focus from training static models to developing agents capable of iterative improvement (Gao et al., 2026). Such agents can autonomously learn from experience and improve their capabilities in an open-ended manner. Existing work studies several self-improvement mechanisms, including reflective reasoning and augmenting behavior with memories from prior interactions (Ouyang et al., 2025; Qian et al., 2024; Liang et al., 2024; Guan et al., 2024). More naturally, self-evolution could also be achieved through updating the training data with agents' interaction with tasks (self-train), where agents generate novel experiences without relying on specific human-provided data (Dou et al., 2024; Guo et al., 2025; Qi et al.; Sun et al., 2025). Notably, self-evolving agents are related to, but distinct from, online RL: rather than requiring continual environment interaction during policy learning, self-evolving agents target practical, rapid policy improvement through iterative updates.

## 3. Preliminary

In this section, we review implicit q learning and behavior proximal policy optimization. We take the following example: learning policy from a dataset $\mathcal{D}$, which consists of trajectories $\tau = \{u\} \cup \{(o_t, a_t, r_{t+1})\}_{t=0}^{T-1}$ with task description $u$, observation $o_t$, action $a_t$, reward $r_{t+1}$ and the historical information $h_t = (u, o_1, a_1, \cdots, o_{t-1}, a_{t-1})$.

**Implicit Q-Learning (IQL)** (Kostrikov et al., 2022) is to learn a critic without explicitly maximizing over out-of-distribution actions. In principle, there are two separate modules in the critic: a max-Q operator surrogate $V(u, h_t, o_t)$

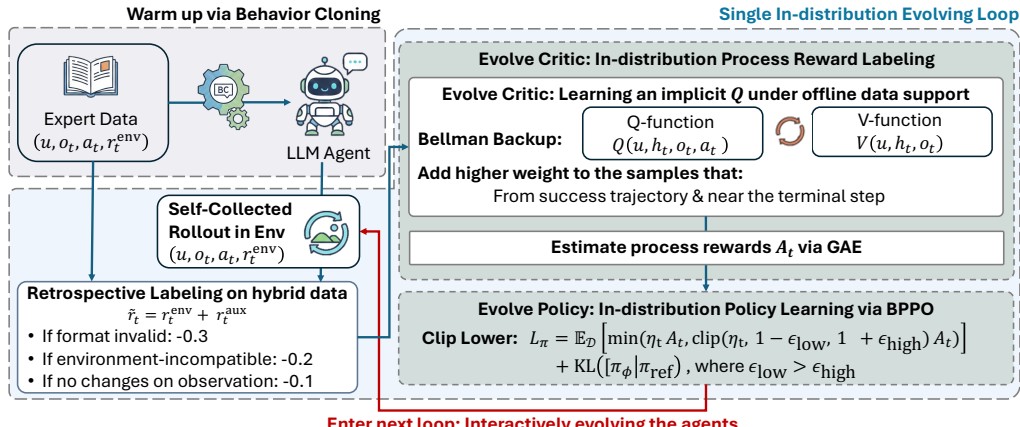

*Figure 2.* Framework of our self-evolving agent. We first warm up the policy via behavior cloning. Then, the agent is iteratively optimized through multiple **in-distribution evolving loops**. In each loop, we construct a hybrid offline buffer by combining expert demonstrations with self-collected trajectories, followed by rule-based retrospective labeling to initialize reward signals. **In-distribution Reward Assignment and Policy Improvement:** Rewards are propagated via Bellman backups to learn a surrogate of the max-Q operator, from which step-level advantages (GAE) are derived. These advantages are further redistributed to the token level to enable provable policy optimization. **Interactive Improvement:** The overall framework forms a closed-loop evolution process, where the policy, critic, and dataset co-evolve, while each update remains constrained within the in-distribution data of each evolving iteration.

that approximates an expectile of the action-value distribution induced by the dataset actions, as well as a Q-function $Q(u, h_t, o_t, a_t)$. Specifically, $V$ is optimized by minimizing an asymmetric regression loss

$$L_V = \mathbb{E}_{\mathcal{D}}\left[L_2^m\big(\bar{Q}(u, h_t, o_t, a_t) - V(u, h_t, o_t)\big)\right], \quad (1)$$

where $m \in (0, 1)$ controls the expectile level, $\bar{Q}$ denotes a slowly-updated target network and $L_2^m(\cdot)$ is the asymmetric squared loss defined as $L_2^m(\delta) = \big|m - \mathbb{1}(\delta < 0)\big|\, \delta^2$. Given $V$, $Q$ is optimized via:

$$L_Q = \mathbb{E}_{\mathcal{D}}\left[\big(r_{t+1} + \gamma V(u, h_{t+1}, o_{t+1}) - Q(u, h_t, o_t, a_t)\big)^2\right]. \quad (2)$$

IQL does not require an explicit behavior policy and only relies on the dataset actions, which help mitigate extrapolation errors in training with offline trajectories.

## 4. Methodology

To address long-horizon delayed rewards without incurring the distribution-shift risks of existing PRM pipelines, our goal is to, given a shared off-policy dataset,

- automatically transform trajectory-level outcomes into stepwise learning signals,
- improve the policy strictly within the same data used for process reward labeling,
- iteratively co-evolve process reward supervision and policy through self-evolving loops, progressively approaching better long-horizon performance.

Crucially, we seek to achieve this through a *self-evolving* learning paradigm, in which the agent iteratively improves itself by (i) generating new experience with its current policy, (ii) re-deriving process-level supervision via an evolving critic, and (iii) updating the policy in a data-constrained manner. This results in a closed-loop evolution process in Q-Evolve, where the policy, critic, and dataset co-evolve, while each policy update remains grounded within the in-distribution hybrid dataset of each iteration.

As shown in Figure 2, we instantiate this paradigm via a data-constrained inner loop that combines process reward acquisition and policy optimization, and further enables self-evolution by periodically refreshing the offline data with newly collected experience.

### 4.1. Behavior Cloning as Policy Warmup

Behavior cloning (BC) offers a simple yet effective approach to initializing agents by replicating expert demonstrations. Define an expert trajectory as $\tau^{\text{exp}} = (u, o_1, a_1, r_2, o_2, a_2, r_3 \ldots, o_{t-1}, a_{t-1}, r_T, o_T)$, where $u$ is the task description, $o_t$ the observation at step $t$, $a_t$ the corresponding expert action including chain-of-thoughts, $r_t$ the environmental rewards, $h_t = (o_1, a_1, \ldots, o_{t-1}, a_{t-1})$ the historical observations and actions up to step $t$. The expert dataset $\mathcal{D}_{\text{expert}}$ is then defined as the collection of expert trajectories. Then we optimize the policy $\pi_\theta$ by minimizing the negative log-likelihood of expert actions conditioned on the history: $\mathcal{L}_{\text{BC}} = -\mathbb{E}_{(u, h_t, a_t) \sim \mathcal{D}_{\text{expert}}}\big[\log \pi_\theta(a_t \mid u, h_t, o_t)\big]$, where $\theta$ denotes the parameters of the LLM agent policy $\pi_\theta$ which output the action in natural language. This objective

encourages the policy to mimic expert demonstrations by reproducing the correct action sequence given the task description and historical information. We denote the policy for this stage as $\pi_{\text{BC}}$.

## 4.2. Data Preparation

Long-horizon interactive tasks feature sparse, delayed rewards and frequent execution errors, making it hard to obtain reliable step-wise supervision from raw trajectories. To enable scalable process-level training signals without additional environment access, we first construct a hybrid offline dataset and then retrospectively relabel each step with auxiliary rewards from textual feedback.

**Hybrid Data Construction.** To automatically obtain informative process-reward labels, we deliberately construct a *mixed* offline dataset by combining expert demonstrations with the agent's own interaction trajectories, as each source resolves a complementary bottleneck in long-horizon learning. Expert data typically contains the key steps and successful subroutines required to solve the task, providing high-quality guidance for both process reward labeling and policy learning. Meanwhile, self-collected experience exposes the policy's actual state-action coverage, including diverse failure modes and locally plausible but wrong actions, so that process supervision is calibrated to where the agent makes mistakes under its true behavior distribution. As a result, we obtain a mix dataset $\mathcal{D} = \mathcal{D}_{\text{expert}} \cup \mathcal{D}_{\text{self}}$, where $\mathcal{D}_{\text{self}}$ is the agents' rollouts in environment, following the self-train paradigm (Dou et al., 2024). In the first policy evolution, $\mathcal{D}_{\text{self}}$ is collected by $\pi_{\text{BC}}$.

**Retrospective Reward Labeling.** A distinctive advantage of LLM-based agents is their interpretable, transparent decision process, where both observations and actions are expressed in natural language, and the environment often provides explicit textual feedback. Inspired by this, we leverage this property to perform *Retrospective Reward Labeling*, i.e., equip each timestep with rule-based auxiliary rewards by parsing the observation feedback to identify invalid actions and meaningless actions. The above procedure does not require access to the environment dynamics, making it practical and broadly applicable.

Specifically, given an offline trajectory $\tau \sim \mathcal{D}$, where $\tau = \{u\} \cup \{(o_t, a_t, r_{t+1})\}_{t=0}^{T-1}$, we retrospectively detect execution failures by (i) validating the format of $a_t$ and (ii) inspecting the subsequent observation $o_{t+1}$, which often explicitly reports invalid actions or execution errors to relabel each step with an auxiliary reward $r_t^{\text{aux}}$,

$$r_t^{\text{aux}} = \begin{cases} r^{\text{fmt}}, & \text{if } o_{t+1} \text{ indicates a format error} \\ r^{\text{inv}}, & \text{if } o_{t+1} \text{ reflecting an invalid action} \\ r^{\text{repeat}}, & \text{if } o_t = o_{t+1} \\ 0, & \text{otherwise} \end{cases} \quad (3)$$

for each timestep. In this way, we provide sparse but fine-grained reward guidance by penalizing non-executable steps immediately, which helps disentangle action validity from task success, promoting the agent to maintain valid environment interactions as much as possible. Please refer to Appendix B.3 for more details.

## 4.3. Estimate Advantage as Process Reward

Below we present how to automatically collect process rewards via advantage estimation, while addressing the challenging critic learning introduced by episodic rewards.

**In-distribution Critic Learning.** Recall our goal is to strictly constrain the process reward labeling and policy learning within a fixed dataset. Therefore, a natural choice is to adopt Implicit Q-Learning (IQL), which explicitly avoids learning over out-of-distribution actions. However, even with principal Bellman Backup in IQL, learning a reliable critic remains challenging under sparse and delayed feedback: Bellman backups can, in principle, propagate terminal supervision backward to earlier steps, but this mechanism often becomes ineffective when episodic rewards are extremely sparse: most transitions receive zero reward, and the learning targets are dominated by noisy bootstrapping. The issue is particularly pronounced for weak agents in early-stage exploration, where trajectories are failure-prone and seldom reach rewarding terminal states, leaving the offline data heavily skewed toward low-signal regions.

To address this, our remedy is threefold, targeting data coverage, learning stability, and signal prioritization. First, we build a hybrid offline dataset that combines expert demonstrations (to cover successful, high-reward regions) with rollouts from weaker policies (to reflect the agent's exploration distribution), providing both informative successes and hard negatives. Second, we decouple critic learning from policy improvement by first training an in-distribution offline critic on the fixed dataset, which avoids a noisy co-learning feedback loop and yields more stable value estimates for downstream supervision. Third, we adopt a simple but efficient weighted IQL objective that prioritizes informative samples by upweighting steps from successful trajectories and placing larger weights on later steps that are more correlated with terminal outcomes.

**Weighted IQL objective.** During critic training, we use a shaped reward $r_{t+1} = r_{t+1}^{\text{env}} + r_{t+1}^{\text{aux}}$, where $r_{t+1}^{\text{env}}$ is the episodic reward returned by the environment and $r_{t+1}^{\text{aux}}$ is assigned by retrospective labeling. We then learn the in-distribution critic with IQL (Eq. 1 and Eq. 2), but reweight each transition to prioritize informative supervision. Concretely, for a trajectory of length $T$, we assign a step weight

$$w_t = (t/T + d) \cdot 0.5 + 0.5, \quad t = 1, \ldots, T, \quad (4)$$

where $d \in \{0, 1\}$ indicates whether the trajectory terminates

with non-zero episodic reward. This design (i) upweights successful trajectories ($d = 1$) and (ii) places larger weights on later steps that are more correlated with terminal outcomes. We incorporate $w_t$ by reweighting the per-transition losses in IQL. For example, the expectile regression becomes

$$L_V = \mathbb{E}\big[w_t \cdot L_2^\tau(\bar{Q}(u, h_t, o_t, a_t) - V(u, h_t, o_t))\big], \quad (5)$$

and we apply the same weighting to the Q-function regression loss. With these weighted objectives, the critic receives stronger signals from informative steps, leading to more stable and reliable value estimates for downstream process-reward labeling and policy improvement.

**Obtain Process Reward Labels**. Given a learned critic, consisting of functions $V$ and $Q$, we use advantage to assess action quality and treat it as a process-level reward signal that reflects long-term returns. To obtain robust advantage estimation, we compute step-wise advantages with generalized advantage estimation (GAE) (Schulman et al., 2016), rather than directly taking the difference of $Q$ and $V$.

For a trajectory $\tau = \{(u, o_t, a_t, r_{t+1})\}_{t=0}^{T-1}$ and $V$ function, we have

$$\begin{aligned}\delta_t &= r_{t+1}^{\mathrm{env}} + \gamma V(u, h_{t+1}, o_{t+1}) - V(u, h_t, o_t), \\ A_t &= \delta_t + \lambda\gamma A_{t+1}, \qquad A_T = 0,\end{aligned} \quad (6)$$

where $r_{t+1}^{\mathrm{env}}$ is environmental episodic rewards.

We find that *excluding* $r^{\mathrm{aux}}$ from advantage estimation yields better performance than either (i) removing $r^{\mathrm{aux}}$ entirely or (ii) also including it in advantage estimation. This design keeps the process reward aligned with the true task objective and preserves the optimal policy invariant; we refer to Appendix A.2 for a formal analysis.

### 4.4. Self-Evolve: Policy Learning with Process Rewards

In order to learn policy from fixed dataset in the inner-loop, our first attempt is to follow IQL and perform advantage-weighted regression (AWR) for policy learning via $L_\pi(\theta) = \mathbb{E}_{(u,h_t,o_t,a_t,A_t)\sim\mathcal{D}}\big[\exp(A_t)\log\pi_\theta(a_t \mid u, h_t, o_t)\big]$. However, we found this formulation prone to overfitting: it monotonically increases the likelihood of actions present in the offline dataset, and provides no mechanism to explicitly *decrease* the probability of actions with negative process-reward signals.

Therefore, we adopt an alternative behavior proximal policy objective (BPPO) that uses the sign and magnitude of process rewards to both upweight beneficial actions and suppress negatively labeled ones.

**In-distribution Policy Optimization.** Similar to learn the critic, our goal is to update the policy over *dataset state and actions only*. Concretely, we use a clipped, behavior-

proximal policy objective:

$$\begin{aligned}\mathcal{L}_\pi(\theta) = \mathbb{E}_\mathcal{D}\Big[&\min\big(\eta_t A_t,\ \mathrm{clip}(\eta_t, 1 - \epsilon_{\mathrm{low}}, 1 + \epsilon_{\mathrm{high}})A_t\big)\Big] \\ &+ \alpha\,\mathrm{KL}(\pi_\phi \mid \pi_{\mathrm{ref}}),\end{aligned} \quad (7)$$

where $\eta_t = \frac{\pi_\phi(a_t|u,h_t,o_t)}{\pi_{\mathrm{old}}(a_t|u,h_t,o_t)}$ is the importance ratio between the current policy and a lagged behavior policy $\pi_{\mathrm{old}}$ used to generate the offline trajectories, $\alpha \in [0, 1]$ controls KL regularization strength, and $\epsilon_{\mathrm{low}}, \epsilon_{\mathrm{high}}$ are hyper-parameters.

*Remark.* Eq. 7 follows the PPO-style clipped surrogate, but differs in how the advantage signal is obtained. Instead of training an online critic with extensive online interactions, we directly use the labeled process rewards $A_t$ introduced above as step-wise advantages for policy updates. Moreover, we use an asymmetric clipping scheme with $\epsilon_{\mathrm{low}} > \epsilon_{\mathrm{high}}$: we permit more aggressive suppression (larger decreases) for negatively labeled actions while keeping probability increases more tightly constrained for preventing overfitting. This encourages conservative, in-support updates that suppress harmful actions without aggressively extrapolating beyond the offline data distribution.

We denote the optimized policy via Eq 7 as $\pi_{\mathrm{evolved}}$.

**Interactive Improvement.** The procedure above defines an *inner loop* that learns an improved policy $\pi_{\mathrm{evolve}}$ from a fixed hybrid dataset. While each inner-loop update is strictly grounded within the in-distribution data of the current policy, the resulting policy can be used to correct the data distribution and unlock further improvements. Concretely, we use $\pi_{\mathrm{evolve}}$ to interact with the environment and collect new trajectories, forming $\mathcal{D}^{\pi_{\mathrm{evolve}}}$, and then refresh the off-policy dataset by merging them with expert demonstrations: $\mathcal{D} \leftarrow \mathcal{D}^{\mathrm{exp}} \cup \mathcal{D}^{\pi_{\mathrm{evolve}}}$. With the updated dataset, we rerun the inner loop to relearn an in-distribution critic and relabel process rewards, yielding a stronger policy for the next iteration. Overall, the pipeline forms a closed-loop evolution process where the policy, critic, and dataset co-evolve, while each policy update remains grounded within the in-distribution hybrid dataset of the current iteration. In our experiments, we run two training loops for SciWorld and AlfWorld, while three loops for WebShop.

The complete Q-Evolve procedure is summarized in Algorithm 1.

## 5. Experiments

Here we empirically evaluate our method and compare it against a range of strong baselines.

### 5.1. Setup

We first describe the experimental setup, including the base model used for agents and the evaluation environments.

*Table 2.* Performance comparison on WebShop, SciWorld (Seen/Unseen), and ALFWorld (Seen/Unseen). The best result is **bolded** and the second-best result is underlined.

| Method | WebShop | SciWorld | | ALFWorld | | Average |
|---|---|---|---|---|---|---|
| | | Seen | Unseen | Seen | Unseen | |
| GPT-4 | 63.2 | 64.8 | 64.4 | 42.9 | 38.1 | 54.7 |
| GPT-3.5-Turbo | 62.4 | 16.5 | 13.0 | 7.9 | 10.5 | 22.1 |
| Reflexion (Shinn et al., 2023) | 64.2 | 60.3 | 64.4 | 45.7 | 55.2 | 58.0 |
| Base Agent (Llama-2-7B-Chat) | 17.9 | 3.8 | 3.1 | 0.0 | 0.0 | 5.0 |
| SFT (Chen et al., 2023) | 63.1 | 67.4 | 53.0 | 60.0 | 67.2 | 62.1 |
| RFT (Zhang et al., 2023b) | 63.6 | 71.6 | 54.3 | 62.9 | 66.4 | 63.8 |
| PPO (Schulman et al., 2017) | 64.2 | 59.4 | 51.7 | 22.1 | 29.1 | 45.3 |
| Best-of-N | 67.9 | 70.2 | 57.6 | 62.1 | 69.4 | 65.4 |
| ETO (Song et al., 2024) | 67.4 | 73.8 | 65.0 | 68.6 | 72.4 | 69.4 |
| DMPO (Shi et al., 2024) | 70.1 | 72.4 | 61.7 | - | - | - |
| QLASS (Lin et al., 2025) | 70.3 | 75.3 | 66.4 | 77.9 | 82.8 | 74.5 |
| **Q-Evolve (Ours)** | **70.5** | **76.3** | **69.7** | **90.7** | **89.6** | **79.4** |

---

**Algorithm 1** Q-Evolve

**Input:** Expert dataset $\mathcal{D}_{\text{expert}}$; environment Env; iterations $K$
**Output:** Evolved policy $\pi_\theta$
Warm up $\pi_\theta$ via behavior cloning on $\mathcal{D}_{\text{expert}}$ **for** $k = 1, \ldots, K$ **do**
  // Stage 1: Hybrid Data Construction
  $\mathcal{D}_{\text{self}} \leftarrow \text{rollout}(\pi_\theta, \text{Env})$;  $\mathcal{D} \leftarrow \mathcal{D}_{\text{expert}} \cup \mathcal{D}_{\text{self}}$  Assign $r_t^{\text{aux}}$ to each step via retrospective labeling (Eq. 3)
  // Stage 2: In-distribution Critic Learning
  Train $V$, $Q$ on $\mathcal{D}$ by minimizing $\mathcal{L}_V$, $\mathcal{L}_Q$ (Eqs. 1–2)
  // Stage 3: Process Reward Derivation
  Compute $A_t$ via GAE on $r^{\text{env}}$ and $V$ (Eq. 6)
  // Stage 4: In-distribution Policy Optimization
  Update $\pi_\theta$ by maximizing $\mathcal{L}_\pi$ (Eq. 7)

**return** $\pi_\theta$

---

**Agent Base Model and Rollout.** We use Llama2-7B-Chat (Touvron et al., 2023) as the base model for building LLM agents following Song et al. (2024); Lin et al. (2025). We list all the prompts used in Appendix B.1. For self-collected data, we sample 3 trajectories per task, while the number of tasks is shown in the Appendix (Table 12).

**Evaluation Tasks** We conduct experiments on three environments that naturally exhibit delayed rewards, ALF-WORLD (Shridhar et al.) for embodied house holding tasks, WEBSHOP (Yao et al., 2022a) for web navigation, and SCI-ENCEWORLD (Wang et al., 2022) for embodied science experiments. ALFWORLD is a text-based embodied environment where agents must complete household tasks through long action sequences, making credit assignment particularly challenging. The agent only receives a binary reward at the final step, 1 for success and 0 for failure. WEBSHOP evaluates goal-oriented dialogue and decision making in an online shopping environment, where rewards are only observed after the action, "click [Buy Now]". If the purchased item satisfies all required attributes, the agent re-

ceives a reward of 1; otherwise, the agent receives a reward proportional to the number of attributes it meets. SCIENCE-WORLD is a text-based virtual environment requiring the agents to complete tasks with subgoals, with sparse success signals from 0 to 1 provided at the end of episodes to indicate the achievement of the subgoals. For ScienceWorld and ALFWorld, we evaluate both seen and unseen tasks to investigate the generalization of agents. We report the average accumulated rewards as the evaluation metrics.

**Baselines.** We compare our method against three categories of baselines: (1) *zero-shot LLMs*, including GPT-3.5-Turbo and GPT-4 with ReAct prompting (Shinn et al., 2023), which are directly applied without task-specific adaptation; (2) *fine-tuned LLMs* trained without reward redistribution, such as **SFT** (Chen et al., 2023), which is supervised fine-tuning on expert trajectories, and **RFT** (Rejection sampling Fine-Tuning) (Zhang et al., 2023b), a self-improvement baseline trained on merged successful trajectories and expert data; (3) *LLMs with existing reward redistribution strategies*, including **ETO** (Song et al., 2024), which updates policies via constructing trajectory-level preference pairs and DPO, **DMPO** (Shi et al., 2024), which utilizes a multi-turn preference objective to optimize the agent, and **PPO** (Schulman et al., 2017), a reinforcement learning baseline optimizing the final reward. In addition, we also evaluate inference-time strategies such as **Best-of-N** (with $N = 6$), **QLASS** (Lin et al., 2025), which constructs an exploration tree to estimate the Q-value of state-action pairs, enabling planning on a behavior cloning agent, and closed-source agents like GPT-4o with **Reflexion** (Shinn et al., 2023).

**5.2. Main Result**

Table 2 shows that our method achieves the best overall performance across all benchmarks, obtaining the highest average score among all baselines. Compared with QLASS,

which similarly relies on value-based signals, Q-Evolve achieves substantially higher scores while reducing dependence on heavy online sampling (600K for QLASS and 20K for Q-Evolve in AlfWorld). In particular, QLASS estimates $Q$-values through online rollouts and search, whereas Q-Evolve learns an in-distribution critic and derives process rewards largely from offline relabeling, enabling more sample-efficient and robust improvements under limited additional interaction. Compared with ETO, Q-Evolve achieves better overall performance, which we attribute to a more stable inner-loop self-evolution that grounds policy updates in a hybrid offline dataset via in-distribution critic learning and process-level supervision. Compared with baselines that do not explicitly address the episodic rewards, Q-Evolve consistently performs better across all tasks, highlighting the benefit and significance that Q-Evolve provides denser and more reliable credit assignment.

## 5.3. Ablation Study

In this subsection, we conduct several ablations to investigate the components in the proposed Q-Evolve. Without specification, we adopt only one interactive policy learning in the ablation study.

**Using process rewards *with* vs. *without* support of the data distribution.** We create an ablation version of w/o PI, using process rewards for out-of-distribution implicit policy learning, where we use the critic to perform test-time scaling. Specifically, we rerank the candidate answers produced by $\pi_{\text{BC}}$, according to the score from the critic, $Q-V$. As shown in Figure 3, empirically, in-distribution policy learning (Full and w/o GAE) outperforms $\pi_{\text{BC}}$. In contrast, using critic test-time scaling does not necessarily help, and can even underperform $\pi_{\text{BC}}$, caused by a distribution mismatch. Such a distribution shift comes from two aspects: first, the models might provide unseen action candidates that may lie outside the PRM's reliable scoring regime; second, the environmental dynamics might push the agent toward unseen states, which are even worse when the policies are implicitly improved. By contrast, our policy training partially mitigates this issue by aligning the policy's learning data distribution with the PRM's distribution and relying on generalizable estimation to estimate a more robust advantage. Therefore, it is crucial to *use process reward labels within a controllable, in-distribution regime where its scoring remains reliable*.

**Ablation on key components.** Table 3 presents a component-wise ablation of our framework on ALFWorld. Overall, removing any single module degrades performance, confirming that the final gains come from their synergy rather than a single trick. First, removing retrospective relabeling (w/o RT) leads to a clear performance drop, indicating that RT provides important intermediate learning signals and improves generalization by identifying obvious failures

*Table 3.* Ablation study on key components on AlfWorld. RR: Retrospective relabeling; W-IQL: Weighted IQL, GAE: generalized advantage estimation and PI: One-step Policy Improvement. w/o PI: using critic in test-time scaling. w/o PI + AWR: using critic with advantage weighted regression for policy improvement.

| Variant | RR | W-IQL | GAE | PI | Seen | Unseen |
|---|---|---|---|---|---|---|
| Q-Evolve (1-iter) | ✓ | ✓ | ✓ | ✓ | **87.9** | **86.6** |
| SFT | ✗ | ✗ | ✗ | ✗ | 60.0 | 67.2 |
| w/o RR | ✗ | ✓ | ✓ | ✓ | 83.6 | 82.7 |
| w/o W-IQL | ✓ | ✗ | ✓ | ✓ | 83.6 | 76.1 |
| w/o GAE | ✓ | ✓ | ✗ | ✓ | 74.3 | 74.6 |
| w/o PI | ✓ | ✓ | ✓ | ✗ | 58.6 | 59.0 |
| w/o PI + AWR | ✓ | ✓ | ✓ | ✓ | 64.3 | 67.9 |

*Table 4.* Comparison of different process-reward choices.

| Process rewards | Seen | Unseen |
|---|---|---|
| $Q - V$ | 74.3 | 74.6 |
| $r^{\text{env}} + \gamma V' - V$ | 80.0 | 74.6 |
| GAE with $r^{\text{env}}$ (Ours, 1-iter) | **87.9** | **86.6** |
| GAE with $r^{\text{env}} + r^{\text{aux}}$ | 81.4 | 82.8 |

without additional environment interaction. Second, **w/o W-IQL**, replacing weighted IQL by standard IQL causes a clear regression, suggesting that weighted IQL improves robustness of the critic under episodic rewards. Third, GAE is a key bridge from critic estimation to policy supervision: removing GAE (w/o GAE) leads to a substantial decline, highlighting the importance of obtaining high-quality advantage estimation. Finally, **one-step policy improvement (PI)** is indispensable for translating step-wise signals into actual policy gains, which we explain with more details in the ablation of in-distribution policy learning *vs.* out-of-distribution policy learning and ablation on the choice of policy learning. Together, these ablations validate that (i) RR, W-IQL, and GAE yield reliable advantage signals, and (ii) PI is the dominant mechanism that converts those signals into consistent improvements.

**Comparison of process reward choice.** We compare multiple alternatives for process rewards: one step advantage $Q - V$, potential-based shaping $r^{\text{env}} + \gamma V' - V$, GAE with $r^{\text{env}}$, and GAE with $r^{\text{env}} + r^{\text{aux}}$, given $Q$ and $V$. As shown in Table 4, GAE with $r^{\text{env}}$ outperforms other alternatives, indicating that multi-step advantage estimation provides more reliable credit assignment while keeping policy updates aligned with task success. In contrast, the one-step $Q - V$ signal is much weaker, likely due to bootstrapping noise in long-horizon tasks. Potential-based shaping $r^{\text{env}} + \gamma V' - V$, suggests that advantage could produce better temporal credit assignment, where $V'$ denotes the value estimation of next timestep. Finally, including $r^{\text{aux}}$ inside GAE hurts overall performance, showing that heuristic auxiliary rewards may bias the policy learning, while still being useful as an

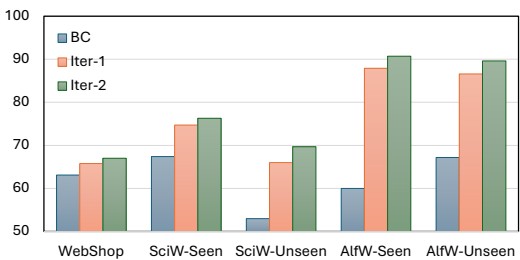

*Figure 3.* Ablation on interactive improvement.

auxiliary objective (Table 3).

**Comparison of policy learning choice: AWR *vs*. BPPO.**
As shown in Table 3 (last row), we construct "w/o PI +
AWR" via replacing the BPPO policy optimization with
advantage-weighted regression, the same as IQL. AWR op-
timizes a weighted behavior cloning objective, where all ac-
tions (including negative-advantage ones) are still imitated,
differing only in their effective learning speed via advantage-
dependent weights. In contrast, our objective uses signed
advantage to explicitly upweight positive-advantage actions
while downweighting negative ones, enabling more direct
correction of harmful behaviors. This ablation highlights
the importance of explicit negative-action suppression in
long-horizon policy improvement.

**Ablation on interactive improvement.** To isolate the effect
of iterative data collection and refinement, we compare two
consecutive rounds of our self-evolving pipeline, shown in
Figure 3. The consistent gain from the first loop (Iter-1) to
the second loop (Iter-2) indicates that each iteration con-
tributes additional useful supervision, and that the pipeline
can stably accumulate improvements across multiple rounds
rather than relying on a one-off boost.

**Sample Efficiency.** Table 5 provides an apples-to-apples
comparison against online RL methods on ALFWorld on
Qwen2.5-7B-Instruct [1] as the backbone, where all online
RL methods are trained for 320K environment steps. Q-
Evolve (1-iter) uses only 13K environment steps, while
outperforming all online RL baselines by a large margin on
both seen and unseen splits. Above shows that, Q-Evolve
has better sample efficiency compared with the alternative
methods.

**Generalization across model architectures** To assess
whether Q-Evolve's gains transfer across model families
and scales, we evaluate on two additional settings. Table 6
reports results with Llama-3-8B-Instruct [2], compared with
some planning-based methods, MPO (Xiong et al., 2025),
KnowAgent (Zhu et al., 2025), WKM (Qiao et al., 2024),

*Table 5.* Sample efficiency comparison on ALFWorld (Qwen2.5-
7B-Instruct). All online RL baselines use 320K environment steps.
**Bold**: best result.

| Method | Env. Steps | Seen | Unseen |
|---|---|---|---|
| PPO (Schulman et al., 2017) | 320K | 59.4 | 67.7 |
| RLOO | 320K | 56.4 | 36.6 |
| GRPO (Feng et al., 2025) | 320K | 39.7 | 32.2 |
| SFT | 0 | 74.9 | 62.3 |
| SFT + PPO | 320K | 72.6 | 77.6 |
| SFT + RLOO | 320K | 75.0 | 51.4 |
| SFT + GRPO | 320K | 66.7 | 74.1 |
| **Q-Evolve (1-iter)** | **13K** | **88.6** | **87.3** |

*Table 6.* Results with Llama-3-8B-Instruct. Best result in **bold**,
second-best underlined.

| Method | WebShop | SciWorld | | ALFWorld | |
|---|---|---|---|---|---|
| | | Seen | Unseen | Seen | Unseen |
| SFT | 63.3 | 65.3 | 57.0 | 79.3 | 80.6 |
| ETO | 68.4 | 81.3 | 74.1 | 77.1 | 76.4 |
| KnowAgent | 64.8 | 81.7 | 69.6 | 80.0 | 74.9 |
| WKM | 66.9 | 82.1 | 76.5 | 77.5 | 78.2 |
| SFT + MPO | 65.5 | 70.2 | 65.9 | 80.7 | 81.3 |
| ETO + MPO | 70.2 | 83.4 | 80.8 | 85.0 | 79.1 |
| **Q-Evolve** | **71.1** | **86.4** | **82.4** | **89.6** | **90.3** |

ETO+MPO. Q-Evolve consistently outperforms all base-
lines across all tasks and both seen/unseen splits, demon-
strating that the method is not tied to any particular model
architecture or initialization.

We also provide an ablation on hyper-parameter $\epsilon_{low}$ and
$\epsilon_{high}$ in Appendix B.4.

# 6. Conclusion

In this work, we introduce Q-Evolve, a self-evolving frame-
work for training LLM agents on long-horizon interactive
tasks under episodic rewards. Our key idea is to derive auto-
matic process-level supervision with in-distribution policy
improvement in an inner loop. Specifically, we learn a critic
from a hybrid offline dataset (expert demonstrations and
agent trajectories) using weighted Implicit Q-Learning. The
value function enables step-wise process-reward labeling via
advantage estimation, yielding dense supervision without
backtracking or human annotation. Guided by these signals,
we adopt behavior-proximal policy optimization to improve
the agent while staying within the in-distribution data of the
policy for each iteration, avoiding distribution shift. This
forms a closed-loop process where the policy, critic, and
dataset co-evolve. Experiments on AlfWorld, WebShop, and
ScienceWorld show consistent gains in sample efficiency,
robustness, and task success.

---

[1]https://huggingface.co/Qwen/Qwen2.5-7B-Instruct

[2]https://huggingface.co/meta-llama/Meta-Llama-3-8B-
Instruct

## Impact Statement

This work introduces a self-evolving framework for LLM agents that can significantly influence the deployment of autonomous systems in complex, real-world environments. By grounding agent evolution in a hybrid in-distribution dataset rather than unconstrained online exploration, our method provides a safer pathway for developing autonomous agents in sensitive domains like robotics and web-based services where trial-and-error costs are high. The elimination of expensive manual step-wise annotations and the need for environment backtracking makes it feasible for smaller organizations to train sophisticated reasoning agents without massive labeling budgets or specialized simulator features. Furthermore, the use of behavior-proximal optimization and KL-regularization ensures that as agents evolve to solve specific tasks, they retain their foundational language capabilities and do not develop harmful, out-of-distribution behaviors. Ultimately, our approach demonstrates that process rewards can be derived automatically from episodic outcomes, offering a scalable alternative to human-in-the-loop supervision, which is essential as AI tasks grow in complexity beyond human monitoring capacities.

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

## Limitations

Q-Evolve has several limitations worth noting. The retrospective reward signals depend on structured environment feedback and may require task-specific adaptation; for instance, the repetition penalty is meaningful only in environments where repeated observations reliably indicate stagnation. The self-evolving loop relies on greedy rollouts for data collection, which reduces trajectory diversity over iterations and can cause the policy to converge to locally optimal but suboptimal behaviors. Finally, while the in-distribution constraint stabilizes learning within each iteration, distribution shift accumulates across iterations as the policy evolves, and the current framework does not explicitly correct for this cross-iteration drift.

## A. Analysis of Design Choices

### A.1. Retrospective reward design.

The auxiliary reward values are set according to three principles that govern their interaction with the primary task signal: (i) *small magnitude* relative to the extrinsic reward, so that policy learning prioritizes task completion rather than penalty avoidance; (ii) *non-dominance in cumulative return*, ensuring that auxiliary signals shape rather than override the episodic reward; and (iii) *severity-based ordering*, where failure types are ranked by their impact on task execution—format errors invalidate the action protocol entirely and carry the highest penalty, non-executable actions violate environmental constraints and receive a moderate penalty, and repeated or no-op actions indicate ineffective exploration with the lowest penalty. These considerations yield $(r^{\mathrm{fmt}}, r^{\mathrm{inv}}, r^{\mathrm{repeat}}) = (-0.3, -0.2, -0.1)$.

Table 7 evaluates the robustness of these choices on ALF-World. Performance is stable across alternative reward scales (**RR-alt**), but deteriorates substantially when the format penalty is inflated to $r^{\mathrm{fmt}} = -1$ (**RR-high-fmt**), as it begins to dominate the cumulative return and undermines the task-level signal. For weighted IQL, both the temporal and success terms contribute positively; the temporal term has a larger effect on unseen tasks, where reliable value propagation is especially important for generalization.

### A.2. Analysis of Retrospective Rewards in Advantage Estimation

We formally justify why $r^{\mathrm{aux}}$ is excluded from the GAE used for policy optimization. Let $V$ be the value function trained under the full shaped reward $r^{\mathrm{env}} + r^{\mathrm{aux}}$, and let $A_t^{\mathrm{env}}$, $A_t^{\mathrm{full}}$ denote the GAE advantages computed with $r^{\mathrm{env}}$ and $r^{\mathrm{env}} + r^{\mathrm{aux}}$, respectively.

**Proposition A.1.** *For all* $0 \le t \le T - 1$,

$$A_t^{full} - A_t^{env} = \sum_{l=0}^{T-t-1} (\gamma\lambda)^l \, r_{t+1+l}^{aux}.$$

*Proof.* The GAE recursion with $A_T = 0$ unrolls to $A_t = \sum_{l=0}^{T-t-1}(\gamma\lambda)^l \delta_{t+l}$, where $\delta_t = r_{t+1} + \gamma V(s_{t+1}) - V(s_t)$. Since $V$ is the same in both cases, the TD residuals differ only by $\delta_t^{\mathrm{full}} - \delta_t^{\mathrm{env}} = r_{t+1}^{\mathrm{aux}}$. Subtracting the two expansions gives the result. □

Proposition A.1 shows that including $r^{\mathrm{aux}}$ in GAE directly adds a discounted auxiliary-return term to the policy-side advantage target. To see why this changes the policy objective, consider first the case $\lambda = 1$: GAE reduces to the Monte Carlo advantage, giving $A_t^{\mathrm{env}} = \sum_{l \geq 0} \gamma^l r_{t+1+l}^{\mathrm{env}} - V(s_t)$ (where the $V$-terms telescope and $V(s_T) = 0$). Using only $r^{\mathrm{env}}$ therefore keeps the policy update aligned with the original environment-return objective $J^{\mathrm{env}}(\pi) = \mathbb{E}_\pi[\sum_t \gamma^t r_t^{\mathrm{env}}]$ and does not change the optimal policy. For $\lambda < 1$, GAE no longer equals the Monte Carlo return exactly, but it still defines an environment-reward $\lambda$-return surrogate; the key identity $A_t^{\mathrm{full}} - A_t^{\mathrm{env}} = \sum_{l=0}^{T-t-1}(\gamma\lambda)^l r_{t+1+l}^{\mathrm{aux}}$ remains exact, and excluding $r^{\mathrm{aux}}$ keeps policy learning aligned with this environment-reward surrogate alone. This is corroborated empirically in Table 4.

### A.3. Weighted IQL design.

The step-weight $w_t$ in Eq. 1 incorporates two complementary terms. The temporal term assigns larger weights to later transitions, which have shorter remaining bootstrap horizons and thus more reliable TD targets under sparse rewards. The success term upweights trajectories with non-zero episodic rewards: under extreme reward sparsity, successful rollouts constitute a small but disproportionately informative subset, and without reweighting, critic learning would be dominated by the abundant low-signal failure trajectories.

### A.4. Memory cost.

Beyond environment interactions, Q-Evolve also incurs lower training and inference overhead: during policy optimization, only the policy and reference models are loaded (two models), matching the memory footprint of GRPO, since the critic's advantages are pre-computed offline and need not be retained. At inference time, Q-Evolve requires no additional critic evaluation, whereas QLASS performs multi-candidate scoring per step, adding non-trivial inference cost.

*Table 7.* Sensitivity analysis on AlfWorld (single iteration). **Bold**: best result.

| Variant | Seen | Unseen |
|---|---|---|
| w/o RR | 83.6 | 82.7 |
| RR-alt $(-0.1, -0.05, -0.03)$ | 85.7 | 84.3 |
| RR-high-fmt $(-1, -0.2, -0.1)$ | 69.3 | 64.2 |
| w/o W-IQL | 83.6 | 76.1 |
| W-IQL w/o temporal term | 83.6 | 79.1 |
| W-IQL w/o success term | 85.0 | 83.6 |
| **Q-Evolve** | **87.9** | **86.6** |

# B. Implementation Details

## B.1. Prompts in Experiments

We list all the prompts used in our experiments in Figure 4 (AlfWorld), Figure 5 (SciWorld) and Figure 6 (Webshop).

## B.2. Critic Model Structure

**Critic Model.** We parameterize the critic with a single pretrained LLM backbone and route computation through lightweight LoRA adapters to predict both the state value $V(s)$ and the state–action value $Q(s, a)$. This shared-backbone design preserves a common representation while enabling head-specific specialization, avoiding the overhead of maintaining separate LLMs for $V$ and $Q$.

Let $f_\theta$ denote the pretrained LLM. Given a tokenized input $x$, it produces hidden states $H = f_\theta(x) \in \mathbb{R}^{B \times T \times d}$. We attach two LoRA adapters: a *value adapter* $\phi_v$ and a *Q adapter* $\phi_q$. For each forward pass, we activate the corresponding adapter:

$$H^{(v)} = f_{\theta, \phi_v}(x), \qquad H^{(q)} = f_{\theta, \phi_q}(x). \qquad (8)$$

The input to the critic is a trajectory formatted as a multi-turn chat transcript (user observations and assistant actions) and then tokenized into a single sequence $x = (x_1, \ldots, x_T)$ using the same chat template as the policy. We tokenize the *full* trajectory once to obtain the token ids, and for each step $t$ we additionally tokenize three *prefixes* of the transcript: (i) the prefix ending at the current observation (state prefix), (ii) the prefix ending after the agent action is appended (state–action prefix), and (iii) the prefix ending at the next observation (next-state prefix). The lengths of these three tokenized prefixes define their end-token indices $p_t^{(s)}, p_t^{(sa)}, p_t^{(s')}$ (i.e., the last token positions of each prefix in the full sequence).

Given hidden states $H^{(v)} = f_{\theta, \phi_v}(x)$ and $H^{(q)} = f_{\theta, \phi_q}(x)$, we represent each segment by the hidden state at its end position (last-token pooling):

$$h_t^{(s)} = H_{p_t^{(s)}}^{(v)}, \quad h_t^{(sa)} = H_{p_t^{(sa)}}^{(q)}, \quad h_t^{(s')} = H_{p_t^{(s')}}^{(v)}.$$

In practice, we can sample multiple steps from the same trajectory (e.g., $K$ steps) and gather the corresponding hidden states in one forward pass; these pooled vectors are then fed into small MLP heads to produce $V(s)$, $V(s')$, and Double-$Q(s, a)$ predictions.

**Prediction heads.** On top of the routed hidden states, we use lightweight heads to produce scalar predictions. For each step, we first pool the routed hidden states at the pre-computed end-token positions, obtaining vectors $h_t^{(s)}, h_t^{(sa)}$, and $h_t^{(s')}$. We then apply one value head $g_v$ and two Q heads $g_{q_1}, g_{q_2}$ for Double Q-learning:

$$
\begin{aligned}
V(s_t) &= g_v\left(h_t^{(s)}\right), \\
Q_1(s_t, a_t) &= g_{q_1}\left(h_t^{(sa)}\right), \\
Q_2(s_t, a_t) &= g_{q_2}\left(h_t^{(sa)}\right).
\end{aligned}
\qquad (9)
$$

**Delayed $Q$ network.** For stable target estimation in Bellman-style backups, we maintain a delayed (target) $Q$ network that mirrors the on-training $Q$ branch. It shares the same backbone $f_\theta$ but uses a separate set of EMA parameters $(\bar{\phi}_q, \bar{g}_{q_1}, \bar{g}_{q_2})$. Concretely, the delayed branch routes the input through the target $Q$ adapter

$$\bar{H}^{(q)} = f_{\theta, \bar{\phi}_q}(x), \qquad (10)$$

pools the end-of-action representation $\bar{h}_t^{(sa)}$ in the same way as the online branch, and predicts target Double-$Q$ values:

$$\bar{Q}_1(s_t, a_t) = \bar{g}_{q_1}\left(\bar{h}_t^{(sa)}\right), \qquad \bar{Q}_2(s_t, a_t) = \bar{g}_{q_2}\left(\bar{h}_t^{(sa)}\right). \qquad (11)$$

The target parameters are updated by an exponential moving average of the online $Q$ parameters:

$$
\begin{aligned}
\bar{\phi}_q &\leftarrow (1 - \lambda_{\text{EMA}}) \bar{\phi}_q + \lambda_{\text{EMA}} \phi_q, \\
\bar{g}_{q_1} &\leftarrow (1 - \lambda_{\text{EMA}}) \bar{g}_{q_1} + \lambda_{\text{EMA}} g_{q_1}, \\
\bar{g}_{q_2} &\leftarrow (1 - \lambda_{\text{EMA}}) \bar{g}_{q_2} + \lambda_{\text{EMA}} g_{q_2}.
\end{aligned}
\qquad (12)
$$

We apply this soft update every $K$ optimization steps; in our implementation, we set $K = 2$ and $\lambda_{\text{EMA}} = 0.005$ for both the $Q$ heads and the $Q$ adapter.

**Optimization with step weights and Double-$Q$.** Each sampled step $t$ is associated with a nonnegative weight $w_t$ (stored in the offline dataset), which upweights more informative steps (e.g., later steps or decisive transitions). We train the critic with *Double-Q*: two separate heads $Q_1$ and $Q_2$ are learned in parallel on the same representation $h_t^{(sa)}$ and the same TD target. Concretely, for a transition $(s_t, a_t, r_t, s_{t+1}, d_t)$ we compute the bootstrap target using the value branch,

$$y_t = r_t + \gamma(1 - d_t) V(s_{t+1}),$$

**Algorithm 2** Critic optimization with Double-$Q$, delayed $Q$ (EMA), and step weights

**Input:** Offline trajectory tokens; per-step indices $p_t^{(s)}, p_t^{(sa)}, p_t^{(s')}$ and weights $w_t$; discount $\gamma$; expectile $m$; EMA rate $\lambda_{\text{EMA}}$; update period $K$.

Initialize online params $(\phi_v, g_v)$ and $(\phi_q, g_{q_1}, g_{q_2})$. Set target params $(\bar{\phi}_q, \bar{g}_{q_1}, \bar{g}_{q_2}) \leftarrow (\phi_q, g_{q_1}, g_{q_2})$

**for** $n = 1, 2, \ldots$ **do**

   Sample a trajectory and select $K$ steps $\{t_k\}_{k=1}^K$ with weights $\{w_{t_k}\}$

   Build a token prefix $x$ truncated to the longest selected next-state prefix

   Compute $H^{(v)} = f_{\theta, \phi_v}(x)$ and $H^{(q)} = f_{\theta, \phi_q}(x)$

   Gather $h_{t_k}^{(s)} = H_{p_{t_k}^{(s)}}^{(v)}$, $h_{t_k}^{(sa)} = H_{p_{t_k}^{(sa)}}^{(q)}$, $h_{t_k}^{(s')} = H_{p_{t_k}^{(s')}}^{(v)}$

   **if** $n$ is a $Q$-update step **then**    // update $Q$

       $V'_{t_k} \leftarrow g_v\left(h_{t_k}^{(s')}\right), y_{t_k} \leftarrow r_{t_k} + \gamma(1 - d_{t_k})V'_{t_k}$

       $Q_{1,t_k} \leftarrow g_{q_1}\left(h_{t_k}^{(sa)}\right), Q_{2,t_k} \leftarrow g_{q_2}\left(h_{t_k}^{(sa)}\right)$

       $\mathcal{L}_Q \leftarrow \sum_{k=1}^K w_{t_k}\left[(Q_{1,t_k} - y_{t_k})^2 + (Q_{2,t_k} - y_{t_k})^2\right]$

       Update $(\phi_q, g_{q_1}, g_{q_2})$ by descending $\nabla \mathcal{L}_Q$

   **else**                             // update $V$

       $V_{t_k} \leftarrow g_v\left(h_{t_k}^{(s)}\right)$

       Compute $\bar{H}^{(q)} = f_{\theta, \bar{\phi}_q}(x)$ and gather $\bar{h}_{t_k}^{(sa)} = \bar{H}_{p_{t_k}^{(sa)}}^{(q)}$

       $\bar{Q}_{1,t_k} \leftarrow \bar{g}_{q_1}\left(\bar{h}_{t_k}^{(sa)}\right), \bar{Q}_{2,t_k} \leftarrow \bar{g}_{q_2}\left(\bar{h}_{t_k}^{(sa)}\right)$

       $\bar{Q}_{t_k} \leftarrow \min(\bar{Q}_{1,t_k}, \bar{Q}_{2,t_k})$

       $\mathcal{L}_V \leftarrow \sum_{k=1}^K w_{t_k} \ell_{\exp}\left(\bar{Q}_{t_k} - V_{t_k}\right)$

       Update $(\phi_v, g_v)$ by descending $\nabla \mathcal{L}_V$

   **if** $n \bmod K = 0$ **then**

       $(\bar{\phi}_q, \bar{g}_{q_1}, \bar{g}_{q_2}) \leftarrow (1 - \lambda_{\text{EMA}})(\bar{\phi}_q, \bar{g}_{q_1}, \bar{g}_{q_2}) + \lambda_{\text{EMA}}(\phi_q, g_{q_1}, g_{q_2})$

and regress both heads to this target with a weighted squared loss:

$$\mathcal{L}_Q = \sum_t w_t \left[(Q_1(s_t, a_t) - y_t)^2 + (Q_2(s_t, a_t) - y_t)^2\right].$$

Using two $Q$ heads helps reduce overestimation and improves stability: when fitting the value function, we form a conservative target from the delayed (EMA) $Q$ network by taking the minimum of the two target heads,

$$\bar{Q}(s_t, a_t) = \min\left(\bar{Q}_1(s_t, a_t), \bar{Q}_2(s_t, a_t)\right),$$

and update $V$ via a (weighted) expectile regression toward $\bar{Q}$:

$$\mathcal{L}_V = \sum_t w_t \ell_{\exp}\left(\bar{Q}(s_t, a_t) - V(s_t)\right).$$

The delayed $Q$ network is not optimized by gradient descent; instead, its adapter and heads are updated by EMA from the on-training $Q$ parameters every $K$ steps.

*Table 8.* Rewards for penalty.

| Task | $r^{\text{fmt}}$ | $r^{\text{inv}}$ | $r^{\text{repeat}}$ |
|---|---|---|---|
| **AlfWorld** | $-0.3$ | $-0.2$ | $-0.1$ |
| **WebShop** | $-0.3$ | $0$ | $0$ |
| **SciWorld** | $-0.3$ | $-0.2$ | $-0.1$ |

*Table 9.* **Hyperparameters.**

| Hyperparameter | Value |
|---|---|
| Batch size | dynamic |
| Number of policy training epochs | 3 |
| Number of critic training epochs | 20 |
| Weight decay | 0.0 |
| Warmup ratio | 0.03 |
| SFT learning rate | 1e-5 |
| LR scheduler type | Cosine |
| Model max length | 4096 |
| Discount factor $\gamma$ | 0.95 |
| Discount factor $\lambda$ in GAE | 0.95 |
| Maximum episode length on WebShop | 10 |
| Maximum episode length on SciWorld | 24 |
| Maximum episode length on ALFWorld | 30 |
| Sampled trajectory number for self-training | 3 |
| Exploration temperature | 2.0 |

### B.3. Hyperparameters

We summarize the hyperparameters used across all stages in this section. All hyperparameters leveraged in our method are in the Table 9 and Table 11.

The hyperparameters of the reward penalty are shown in Table 8. In general, we impose a stronger penalty on action–think format errors than on invalid-but-parsable actions, since the former violates the interaction protocol and typically prevents any meaningful environment transition, whereas the latter reflects a semantically infeasible choice under a valid protocol. We additionally penalize non-meaningful actions when the observation remains unchanged after executing an action, which encourages exploration and discourages ineffective interactions; we exclude cases where no-op behavior is part of the environment de-

*Table 10.* Ablation study on hyper-parameters $\epsilon_{\text{low}}$ and $\epsilon_{\text{high}}$ on AlfWorld. Best in each split is in **bold**.

| $\epsilon_{\text{low}}$ | $\epsilon_{\text{high}}$ | Seen | Unseen |
|---|---|---|---|
| 0.2 | 0.2 | 77.9 | 78.4 |
| 0.4 | 0.2 | 74.3 | 77.6 |
| 0.2 | 0.4 | 75.7 | 81.3 |
| 0.4 | 0.4 | 80.7 | 85.1 |
| 0.8 | 0.4 | **87.9** | **86.6** |
| 0.4 | 0.8 | 81.4 | 83.6 |
| 0.8 | 0.8 | 86.4 | 85.8 |

*Table 11.* Hyper-parameters for Iteration 1 vs Iteration 2.

| Hyper-parameter | Iter 1 | Iter 2 |
|---|---|---|
| Environment | `AlfWorld`/`SciWorld`/`WebShop` | `AlfWorld`/`SciWorld`/`WebShop` |
| Token length (max) | 3600/4096/4096 | 3600/4096/4096 |
| Learning rate for Policy Learning | 1e-5/5e-6/5e-6 | 5e-6/5e-6/5e-6 |
| Learning rate for Critic Learning | 1e-5/1e-5/1e-5 | 1e-5/5e-6/5e-6 |
| Epochs for Policy Learning | 3/2/2 | 2/2/2 |
| Epochs for Critic Learning | 20/20/20 | 2/2/2 |
| Steps for old policy backup | no/50/200 | 200/200/200 |

*Table 12.* Dataset statistics. "Test (Seen)" and "Test (Unseen)" indicate evaluation scenarios. "Ave. Steps" is the average interaction steps.

| Task | Train | Test (Seen) | Test (Unseen) | Ave. Steps |
|---|---|---|---|---|
| WebShop | 1938 | 200 | - | 4.9 |
| ScienceWorld | 1483 | 194 | 241 | 14.4 |
| ALFWorld | 3321 | 140 | 134 | 10.1 |

sign (e.g., the `wait` action in SciWorld, and unchanged observations in WebShop).

### B.4. Additional Experiments.

**Ablation study on Hyper-parameter $\epsilon_{\text{low}}$ and $\epsilon_{\text{high}}$ on Alfworld.** As shown in Table 10, larger clipping thresholds allow more aggressive policy updates. When $\epsilon$ is small, varying $\epsilon_{\text{low}}$ and $\epsilon_{\text{high}}$ has only a mild effect, since the update magnitude is tightly constrained. As $\epsilon$ increases, using a larger $\epsilon_{\text{low}}$ becomes more beneficial: it enables stronger down-weighting of known incorrect actions, which reduces overfitting to suboptimal behaviors and leads to better generalization on the Unseen split.

### B.5. Runtime Analysis

All experiments were conducted on a single node with 4×H100 GPUs. In our pipeline, behavior cloning takes approximately 5 minutes per epoch, critic training takes about 1 hour per epoch, and policy learning takes around 15 minutes per epoch. Overall, the runtime is dominated by critic training, while the policy learning and behavior cloning stages are comparatively lightweight.

Interact with a household to solve a task. Imagine you are an intelligent agent in a household environment and your target is to perform actions to complete the task goal. At the beginning of your interactions, you will be given the detailed description of the current environment and your goal to accomplish.

For each of your turns, you will be given the observation from the last turn. You should first think about the current condition and plan your future actions, and then output your action in this turn. Your output must strictly follow this format:
Thought: your thoughts.
Action: your next action.

The available actions are:

1. go to {`recep`}
2. take {`obj`} from {`recep`}
3. put {`obj`} in/on {`recep`}
4. open {`recep`}
5. close {`recep`}
6. toggle {`obj`} {`recep`}
7. clean {`obj`} with {`recep`}
8. heat {`obj`} with {`recep`}
9. cool {`obj`} with {`recep`}

where {`obj`} and {`recep`} correspond to objects and receptacles.
After each turn, the environment will give you immediate feedback based on which you plan your next few steps. If the environment outputs `"Nothing happened"`, that means the previous action is invalid and you should try more options.

Your response should use the following format:
Thought: <your thoughts>
Action: <your next action>

*Figure 4.* The instruction prompt provided to the language agent on AlfWorld.

You are a helpful assistant to do some scientific experiment in an environment.
In the environment, there are several rooms: kitchen, foundry, workshop, bathroom, outside, living room, bedroom, greenhouse, art studio, hallway
You should explore the environment and find the items you need to complete the experiment.
You can teleport to any room in one step.
All containers in the environment have already been opened, you can directly get items from the containers.

The available actions are:
open OBJ: open a container
close OBJ: close a container
activate OBJ: activate a device
deactivate OBJ: deactivate a device
connect OBJ to OBJ: connect electrical components
disconnect OBJ: disconnect electrical components
use OBJ [on OBJ]: use a device/item
look around: describe the current room
examine OBJ: describe an object in detail
look at OBJ: describe a container's contents
read OBJ: read a note or book
move OBJ to OBJ: move an object to a container
pick up OBJ: move an object to the inventory
pour OBJ into OBJ: pour a liquid into a container
mix OBJ: chemically mix a container
teleport to LOC: teleport to a specific room
focus on OBJ: signal intent on a task object
wait: task no action for 10 steps
wait1: task no action for a step
Your response should use the following format:

Thought: `<your thoughts>`
Action: `<your next action>`

*Figure 5.* The instruction prompt provided to the language agent on SciWorld.

You are web shopping.
I will give you instructions about what to do.
You have to follow the instructions.
Every round I will give you an observation and alist of available actions, you have to respond anaction based on the state and instruction.
You can use search action if search is available.
You can click one of the buttons in clickables.
An action should be of the following structure:
**search[keywords]**
**click[value]**
If the action is not valid, perform nothing.Keywords in search are up to you, but the valuein click must be a value in the list of available actions.Remember that your keywords in search shouldbe carefully designed.Your response should use the following format:Thought: I think ...Action: click[something]

*Figure 6.* The instruction prompt provided to language agent on WebShop.

