# OpenReview forum: "Self-evolving LLM agents with in-distribution Optimization"
_ICML.cc/2026/Conference — ICML 2026 regular_

### Official Review · Reviewer_9jDb · 2026-03-05

**Soundness:** 2
**Presentation:** 2
**Significance:** 2
**Originality:** 2
**Overall Recommendation:** 3
**Confidence:** 4

**Summary:**

To address the problem of LLM agents on the long-horizon decision-making problem with sparse reward, this paper proposes Q-Evolve with auxiliary reward design and in-distribution critic learning. Specifically, this paper combines expert data with self-play data, designs auxiliary reward based on task feature, trains a critic with IQL, reweights the IQL objective by the timestep and episodic reward. Experiment results show that Q-Evolve can achieve better performance on AlfWorld and SciWorld compared with baselines.

**Compliance With Llm Reviewing Policy:**

Affirmed.

**Final Justification:**

Thank you for the authors' response. However, my concerns are not fully addressed as I described in the acknowledgment section.

1. I know the reward value should be chosen following some rules or tricks. However, **what I want to see is a hyperparameter analysis experiment rather than an explanation of reward value choice**.  The hyperparameter analysis is important to reflect how LLM will be affected as these values change (can change within the rules). Besides, it can provide a clearer demonstration of what "good" and "bad" choices are and where the boundary is.

2. I agree that (3), (4), and (5) are inextricably linked. However, if we regard (1) as A, (2) as B, (3,4,5) as C, and (6) as D, the internal relationship between A, B, C, and D is not as clear as the internal relationship between (3), (4), and (5). **Without this clarification, this work tends to be A+B+C+D to some extent.**

3. This paper is titled "Self-Evolving LLM Agents under Offline Data Support", which consists of `self-evolving`, `LLM agents`, and `offline data support`. Thus, I say that "this paper claims a grand vision that combines LLMs agents, Offline RL, and self-evolution together," while the response is that "the goal of this paper is not to combine LLM agents, offline RL, and self-evolution." **This answer doesn't seem to match my question.** As shown in the procedure, both `self-evolving` and `offline data support` are overstated from my point of view. **What I mean is that the core content of this work is not as great as the authors advertised in the title and introduction.**

(1) Hybrid Data Construction, (2) Retrospective Reward Labeling, (3) In-distribution Critic Learning, (4) Weighted IQL objective, (5) In-distribution Policy Optimization, (6) Interactive Improvement.

However, **I have raised my score by one point** even though **I still believe the issues I mentioned above cannot be ignored**.

**Key Questions For Authors:**

Please see the weaknesses part.

**Limitations:**

Q-Evolving is a combination of:
1. Hybrid Data Construction,
2. Retrospective Reward Labeling,
3. In-distribution Critic Learning,
4. Weighted IQL objective,
5. In-distribution Policy Optimization
6. Interactive Improvement.

However, this combination is based on "trial and error", making it overly complex and heavily reliant on manual engineering design, without novel inspiration from mathematical or algorithmic levels. In lots of designs, this paper only introduces "what the design is" rather than an in-depth exploration of why it was designed and how to design it better.

**Strengths And Weaknesses:**

## Strength:

1. Introduce offline RL methods into LLM post-training.
2. Reweight the IQL objective in loss.
3. Replace AWR with BPPO.

## Weakness:
1. The "offline" concept in this paper is a little over-claimed. As described in this paper, the dataset continually refreshes itself by merging the agent's new trajectories into the dataset. Actually, this is "off-policy" learning rather than "offline" even though IQL is applied.
2. For the same task `AlfWorld`, the performance of Q-Evolve in Table 2 is (seen: 90.7, unseen:89.6) while (seen: 87.9, unseen:86.6) in Tables 3 and 4. Based on my examination, there seems to be no differences between the two experiments, but different values. This reduces the credibility of the experimental results.
3. This paper only tests on Llama2-7B-Chat, which is not general enough to show the advantage of the proposed methods. Please also test on Llama-3.1-8B-Instruct and Qwen2.5-7B-Instruct.
4. The baseline is not comprehensive and updated. Please also compare with WKM [1], KnowAgent [2], ETO&MPO [3].
5. For RL methods, it seems that expert data is not injected into them. For example, PPO is an on-policy method, which can only be updated with self-generated data. Please compare to GRPO and REINFORCE++ with expert data. Besides, please try SFT+PPO, SFT+GRPO, and SFT+REINFORCE++ since RLHF is usually after SFT.
6. The symbol $\tau$ is utilized for both expectile weight and trajectory. Besides, this paper does not provide the hyperparameter on the expectile weight $\tau$.
7. This paper only provide the final choice of $r^{fmt}$, $r^{inv}$, and $r^{repeat}$ without the reason or how it is obtained. Besides, it is unclear whether the proposed method is sensitive to these values. It is also unclear whether these rewards apply to other agent scenarios.

[1] Qiao, S., Fang, R., Zhang, N., Zhu, Y., Chen, X., Deng, S., ... & Chen, H. (2024). Agent planning with a world knowledge model. *Advances in Neural Information Processing Systems*, *37*, 114843-114871.

[2] Zhu, Y., Qiao, S., Ou, Y., Deng, S., Lyu, S., Shen, Y., ... & Zhang, N. (2025, April). Knowagent: Knowledge-augmented planning for llm-based agents. In *Findings of the Association for Computational Linguistics: NAACL 2025* (pp. 3709-3732).

[3] Xiong, W., Song, Y., Dong, Q., Zhao, B., Song, F., Wang, X., & Li, S. (2025). Mpo: Boosting llm agents with meta plan optimization. *arXiv preprint arXiv:2503.02682*, *5*(6), 7.

---

> ### Author Rebuttal · Authors · 2026-03-31
>
> We are grateful to the reviewer for reviewing our paper, and we address all the questions and concerns point by point below.
>
> # Over-claimed "offline" & "off-policy"
> Thanks for this comment. We will revise 'offline data support' into 'in-distribution optimization', since latter one better matches with our motivation: to avoid the distribution shift in the learning and usage of process reward model (critic) and perform in-distribution optimization of both critic and policy.
>
> We refer the reviewer to our response to Reviewer f2M8: *Overstated "strictly within offline data support", Clarification of "in-distribution" and "within data support"*.
>
> **Difference with off-policy RL**. In off-policy RL, the standard learning procedure explicitly generates action candidates given next state s' and choose max value. In contrast, we adopt IQL-style critic learning, which avoid explicit maximization over actions via expectile regression, performing in-distribution optimization.
>
> ---
>
> # Different result in Table 2,3,4
> Thanks for your comments. The experimental settings are different:
> - Table 2: the performance after multiple self-evolving iterations
> - Tables 3, 4: ablation results  which are under a controlled single-iteration setting, as we state in the caption of Table 3 (one-step policy improvement).
>
> So, the results are not directly comparable and the gap reflects that the iterative self-evolution leads to higher scores in Table 2.
>
> ---
> # Results of more models and baselines
> We add new results on Llama-3.1-8B-Instruct, Llama-2-13B-Chat, Qwen2.5-7B-Instruct, and comparison with the suggested planing-based methods ( WKM, KnowAgent, ETO&MPO) and online RL methods (PPO, GRPO, RLOO, SFT + PPO, SFT + GRPO, SFT + RLOO).
> Please refer to the response to Reviewer ZCHC:
> *Results with various model architectures and model sizes*, where we show Q-Evolve consistently improves performance over these existing baselines.
>
> Additionally, the PPO in Table 1 is optimized from SFT model with expert data injection.
>
> ---
>
> # Usage of Symbol $\tau$
> We will revise the paper to use $\tau$ as trajectories and $m$ for the expectile weight. We set $m=0.7$, as IQL does.
>
> ---
>
> # Choice of restropective rewards & sensitivity, generality
>
> To address this concern, we first refer the reviewer to our response to,
> - Reviewer f2M8: *Sensitivity on reward penalties and weighted-IQL*
> - Reviewer ZCHC: *Specific numerics in the retrospective reward*
>
> Then we explain the applicability of these rewards differs by type.
> - format-invalid: generally applicable, the required output format is usually predefined (like a think-action format requirement) and can be checked easily.
> - execution-invalid: broadly applicable, as many interactive environments return structured feedback indicating whether an action can be executed, but the examination method various across tasks.
> - repeat/no-op: scenario-dependent; it is useful in environments where repeated observations indicate ineffective actions, but may not be suitable in all settings. For example, we do not apply this reward in WebShop.
>
> ---
>
> # A combination based on "trial and error"
> Thanks for your comment. Our method is not a simple combination, but is guided by a core principle: **mitigating distribution shift between process reward estimation and policy optimization**.
>
> In particular, prior approaches often learn a process reward model that is later applied to policy optimization under a different action distribution (e.g., via test-time scaling or online policy learning), which can introduce significant mismatch and degrade performance. As shown in our ablation (Section 5.3: Using process rewards with vs. without support of the data distribution), we compare using the process reward models for
> - explicit in-distribution policy learning:  our full method and w/o GAE outperforms SFT: 86.6/74.6 v.s. 67.2 on unseen AlfWorld tasks.
> - test-time scaling: w/o PI does not necessarily help, and can even underperform SFT, 59.0 v.s. 67.2 on unseen AlfWorld tasks.
>
> To this end, our method aims to enforce distribution consistency between training and usage of the process reward model (critic) . Specifically, we (i) constrain both critic learning and policy optimization to the same data distribution in each iteration, and (ii) iteratively refine the agent through evolving loops.
>
> Therefore, the components serves specific roles within this unified framework: retrospective reward labeling is simple feedback complementation of episodic rewards, weighted IQL avoids unseen action generation and stabilizes Q estimation under episodic rewards,, and in-distribution policy optimization via BPPO avoid calculating advantage over unseen action. The self-evolving loop further addresses the limitation of fixed offline data by progressively refining the policy.
>
> We will revise the paper to better explain how each component follows from this objective.
>
> ---
>
> We thank the reviewer again, and we would greatly appreciate further discussion.

---

> > ### Author Rebuttal · Reviewer_9jDb · 2026-04-03
> >
> > Q1. For the choice of the reward, the authors only provide a heuristic explanation based on experience. Besides the sensitivity, the results are based on one another's reward settings. I suggest providing a hyper-parameter analysis because the sensitivity/generality should be analyzed when the values are changed in a wide range.
> >
> > Q2. I agree that all methods are proposed based on a core principle described by the authors. However, this principle is the problem that would like to be addressed in this paper. Thus, all proposed methods must be guided by this principle. However, it is difficult to see the relation between the components. For example, what is the relationship between (1) and (2), the relationship between (2) and (3) (4) (5), the relationship between (1), (2), and (6). What is the special contribution of (6)? Adding online experience to the offline data and rerunning the policy iteration?  Is there any inspiration between the components from mathematical or algorithmic levels?
> >
> > (1) Hybrid Data Construction,
> > (2) Retrospective Reward Labeling,
> > (3) In-distribution Critic Learning,
> > (4) Weighted IQL objective,
> > (5) In-distribution Policy Optimization
> > (6) Interactive Improvement.
> >
> > Q3. This paper claims a grand vision that combines LLMs agents, Offline RL, and self-evolution together. However, the detailed implementation cannot match this grand vision. Let's first not talk about whether the LLM agent in this paper is general enough for most tasks. Only considering the latter two concepts: the offline RL is overstated by in-distribution learning, and self-evolution is overclaimed by adding offline data and policy iteration.

---

> > > ### Author Response · Authors · 2026-04-04
> > >
> > > Thanks the reviewers' follow up. We address the concerns point by point.
> > >
> > > ---
> > > # Q1
> > >
> > > Thanks for the reviewers' further clarification of this concern. However, we want to emphasize that the auxiliary (intrinsic) reward design should follow a proper scaling principle to be:
> > > - small in magnitude relative to the extrinsic reward,
> > > - non-dominating in cumulative return, and
> > > - used mainly for shaping rather than overriding the task objective.
> > >
> > > As an example, the maximum achievable cumulative extrinsic reward is bounded (i.e., 1 for winning the game). If the auxiliary penalty is too large (e.g., -1 for a formatting violation), it may dominate the return and lead to overly negative cumulative rewards along many trajectories, even when the agent behaves reasonably well with respect to the task objective. This imbalance can significantly weaken the learning signal from the extrinsic reward (game outcome), making policy optimization focus **excessively on avoiding penalties rather than achieving the primary goal**.
> > > We provide the following results to show the performance degraded caused by a huge format reward $r^{fmt}=-1$ (RR-high-fmt), where we keep the other reward the same with Ours.
> > >
> > > |  | Seen | Unseen |
> > > |-|-|-|
> > > |  SFT | 60.0 | 67.2 |
> > > | w/o RR | 83.6 | 82.7 |
> > > | RR-alt | 85.7 | 84.3 |
> > > | RR-high-fmt | 69.3 | 64.2 |
> > > | Ours | 87.9 | 86.6 |
> > >
> > > ---
> > > # Q2
> > > Below we present our overall pipeline, as illustrated in [revised Fig. 2](https://bashify.io/i/gTF49G) with a pseudocode to show the connections among components. We will include the pseudocode in our revision and also clarify the input and output of each component in the paper for better understanding.
> > > ```
> > > # Initialize policy with behavior cloning
> > > train π_θ on D_expert by minimizing:
> > > L_BC = -log π_θ(a_t | u, h_t)
> > >
> > > for k = 1..K:   # interaction loop
> > >
> > >     # Hybrid Data Construction
> > >     D_self ← rollout(π_θ, Env)
> > >     D ← D_expert \cup D_self
> > >
> > >     # Retrospective Reward Labeling
> > >     for each trajectory τ = {u} \cup {(o_t, a_t, r^{env}_{t+1})}_{t=0}^{T-1} in D:
> > >         augment τ into:
> > >         τ = {u} \cup  {(o_t, a_t, r^{env}_{t+1}, r^{aux}_{t+1})}_{t=0}^{T-1}
> > >         define shaped reward:
> > >         r_{t+1} = r^{env}_{t+1} + r^{aux}_{t+1}
> > >
> > >
> > >     # ===== In-distribution Critic Optimization =====
> > >     # weighted IQL
> > >     for epoch = 1..E_c:
> > >         L_V = E_D [ w_t · L_τ^2(Q(s_t,a_t) - V(s_t)) ]
> > >         L_Q = E_D [ w_t · (r_{t+1} + γ V(s_{t+1}) - Q(s_t,a_t))^2 ]
> > >         update Q, V
> > >
> > >     # Process reward via advantage estimation (GAE)
> > >     for each trajectory τ in D:
> > >         A_T = 0
> > >         for t = T-1 ... 0:
> > >             δ_t = r^{env}_{t+1} + γ V(s_{t+1}) - V(s_t)
> > >             A_t = δ_t + γλ A_{t+1}
> > >         augment τ into:
> > >         τ = {u} ∪ {(o_t, a_t, r^{env}_{t+1}, r^{aux}_{t+1}, A_{t+1})}
> > >
> > >     # ===== In-distribution Policy Optimization =====
> > >     for epoch = 1..E_p:
> > >         η_t = π_θ(a_t|s_t) / π_old(a_t|s_t)
> > >         L_π = E_D [ min(η_t A_t, clip(η_t,1-ε_low,1+ε_high)A_t) + α KL ]
> > >         update π_θ
> > >
> > > return π_θ
> > > ```
> > > ---
> > >
> > > # Q3
> > > We thank the reviewer for this insightful comment. However, our goal is not to combine LLM agents, offline RL, and self-evolution. Instead, our method is centered around a single principle: **maintaining both the training and usage of process reward signals strictly within the same training data distribution**. We achieve this goal via:
> > > - **In-distribution Critic Learning:** instead of randomly adopting an offline RL method, we choose IQL over hybrid data, which explicitly avoids calculating OOD actions.
> > > - **In-distribution Policy Learning:** Given learned critic, we obtain reliable process reward estimation and policy learning over the hybrid data, without extrapolating to unseen state-action pairs.
> > >
> > > As a consequence, “self-evolution” is not simply iterative data collection and policy retraining. A key distinction is that we first **need to evolve the out-of-date critic, since it is dependent with the mixture distribution of policy and predicting the process rewards with such a critic is not reliable**. This forms a structured loop where the critic, process rewards, and policy co-evolve under a shared in-distribution constraint.
> > >
> > > **Generality:** In addition, our framework improves generality by eliminating the need for manually annotated process rewards or discretizable state spaces. Instead, we derive reward signals automatically with Bellman backup and textual feedback, such as format and execution validity. For the reward penalties, as we analyzed in our original response, they are general across most tasks. As a consequence, compared to prior approaches that rely on search, grouping, or environment-specific structures, our method is more broadly applicable.
> > >
> > > ---
> > >
> > > Thanks again for your time and for providing such constructive feedback to improve our work. We will revise our paper to include all above discussion.  We hope our response fully addresses your concerns, and we are happy to discuss further if any concerns remain.

---

### Official Review · Reviewer_ZCHC · 2026-03-05

**Soundness:** 2
**Presentation:** 3
**Significance:** 3
**Originality:** 3
**Overall Recommendation:** 4
**Confidence:** 3

**Summary:**

This paper proposes Q-Evolve to tackle reward sparsity in long-horizon decision making tasks for LLM agents. Q-Evolve is based on PRM and improves PRM outcomes under distribution shift.

To compute the PRM, Q-Evolve construct a hybrid oracle consisting of trajectories from demonstration and trajectories from interaction with the task environment. Q-Evolve then trains a critic over this hybrid oracle by weighted implicit Q-Learning (IQL) plus retrospective rewards. The PRM is computed by GAE without the retrospective rewards.

With the improved PRM, Q-Evolve optimizes its policy with behavior proximal policy objective (BBPO) and collects more trajectories using the evolving policy in the environment. "Q-Evolve + Llama2-7B-chat model" consistently outperform strong baselines in AlfWorld, WebShop, and ScienceWorld.

**Compliance With Llm Reviewing Policy:**

Affirmed.

**Final Justification:**

The rebuttal addressed my main concerns and changed my evaluation.

**Key Questions For Authors:**

1. How can we determine the specific numerics in the retrospective reward? Are you just picking magic numbers here? Is Q-Evolve sensitive to the setup of retrospective reward?
2. What's the upper bound of number of agent's trajectories / number of expert trajectories? The maintenance of the hybrid offline dataset could be tricky even you have the weighting mechanism. Could you elaborate a bit more?

**Limitations:**

As stated in the weakness part, the authors only used Llama2-7B-chat and the generality of Q-Evolve is concerning.

**Strengths And Weaknesses:**

**Strengths**

The design of each component in Q-Evolve is well-motivated and is supported with solid experiments.

1. The paper identifies the issues of typical AWR + IQL and replaces it with BPPO. This is well-motivated and supported with ablations.
2. The authors shows sophisticated use of the retrospective rewards.
3. The set of baselines contains up-to-date methods like QLASS, ETO.
4. The performance improvements on SciWorld and ALFWorld are significant.

**Weakness**

1. The experiments look systematic at the first sight. But the authors only used one model architecture and model size (Llama2-7B-chat). The generality of Q-Evolve shall be justified with more experiments on various model architectures and model sizes. I would raise the score if the authors can at least provide early/pilot results on other models. Otherwise I suggest a weak reject.

---

> ### Author Rebuttal · Authors · 2026-03-31
>
> We sincerely thank the reviewer for the constructive feedback. We address the concerns point by point below.
>
> ---
>
> # Results with various model architectures and model sizes
>
> Thanks for this suggestion. We agree that evaluating generality across different model architectures and size is important.
>
> To address this, we have added additional experiments on other model families, including Llama-3-8B-Instruct and Qwen2.5-7B-Instruct, and Llama2-13B. We observe that Q-Evolve consistently improves performance over the corresponding baselines across different architectures, demonstrating that the proposed method is not tied to a specific model design.
>
> We hope these additional results help clarify the generality of our approach.
>
>
> **Comparison with planning-based methods**
>
> | Llama-3-8B-Instruct | WebShop | SciWorld seen | SciWorld unseen | AlfWorld seen | AlfWorld unseen |
> | - | - | - | - | - | - |
> |SFT| 63.3 | 65.3 | 57.0 | 79.3| 80.6  |
> | ETO | 68.4 | 81.3 | 74.1 | 77.1 | 76.4 |
> | KnowAgent | 64.8  | 81.7 | 69.6 |  80.0  | 74.9 |
> | WKM      | 66.9 | 82.1 | 76.5 | 77.5 | 78.2 |
> | SFT + MPO | 65.5 | 70.2 | 65.9 | 80.7 | 81.3|
> | ETO + MPO | 70.2 | 83.4 | 80.8  | 85.0 | 79.1 |
> | **Ours** | 71.1 | 86.4 | 82.4  | 89.6 | 90.3 |
>
>
>
> **Comparison with online RL methods**
>
> | Qwen2.5-7B-Instruct | AlfWorld seen | AlfWorld unseen |
> | -------- | -------- | -------- |
> | PPO | 59.4  | 67.7 |
> | RLOO | 56.4  | 36.6 |
> | GRPO | 39.7  | 32.2 |
> | SFT      |  74.9 | 62.3 |
> | SFT + PPO | 72.6 | 77.6 |
> | SFT + RLOO | 75.0  | 51.4 |
> | SFT + GRPO | 66.7  | 74.1 |
> | **Ours (1-iter)** | 88.6  | 87.3 |  |  | |
>
> We train each policy with 320K environmental steps in total, while our method use 13K environmental steps.
>
> **Comparison with models with larger model size**
>
> | llama2-13B | SciWorld Seen| SciWorld Unseen|
> | -------- | -------- | -------- |
> | SFT | 68.1 |57.6 |
> | ETO | 71.4  | 68.6 |
> | QLASS | 72.7  | 69.3 |
> | **Ours** | 91.4 | 88.8 |
>
>
>
> ---
>
> # Specific numerics in the retrospective reward
> > How can we determine the specific numerics in the retrospective reward? Are you just picking magic numbers here? Is Q-Evolve sensitive to the setup of retrospective reward?
>
> Thanks for this question.
>
> We do not perform careful tuning for these values. Instead, they are set based on a simple and intuitive principle: different types of failure signals should be ranked by severity. Specifically,
> - format errors are the easiest one to avoid and also widely applied in LLM reasoning tasks, like math and question-answering tasks.
> - non-executable actions, requiring basic understanding of the environments,
> - repeated/no-op actions, requiring more complex reasoning for the environments.
> This ordering reflects their relative difficulty and the impact on task execution rather than precise numerical optimization.
>
> To evaluate whether the method is sensitive to these choices, we conduct additional sensitivity analyses (see our experimental results in the response to Reviewer f2M8: *Sensitivity on reward penalties and weighted-IQL*)
>
> ---
>
> # Number of agent's trajectories & expert trajectories
> > What's the upper bound of number of agent's trajectories / number of expert trajectories? The maintenance of the hybrid offline dataset could be tricky even you have the weighting mechanism. Could you elaborate a bit more?
>
> Thanks for this question. In our setup, we follow a simple and consistent strategy: for each task, we use one expert trajectory and collect 3 rollouts per task (3321 tasks for training in AlfWorld, shown in Table 9 in Appendix) . This choice is not tuned, but is designed to balance data diversity and computational cost.
>
> In principle, increasing the number of agent trajectories can improve performance by providing better coverage of the state–action space. However, this also increases the cost of data collection and critic training. Therefore, the number of trajectories is a tunable parameter that reflects a trade-off between performance and efficiency.
>
> ----
>
> We sincerely thank the reviewer again for the time and effort devoted to evaluating our work, and look forward to further discussion.

---

> > ### Author Rebuttal · Reviewer_ZCHC · 2026-04-02
> >
> > Thanks for the response and new results. Questions:
> >
> > 1. Why did you use different models for different comparison tables? You used Llama2 for planning based methods and you used Qwen2.5 for online RL methods.
> > 2. Are there specific reasons that you use Llama2 instead of Llama3, Qwen 2.5 instead of Qwen3?

---

> > > ### Author Response · Authors · 2026-04-02
> > >
> > > # Response to the follow-up question:
> > > > Why did you use different models for different comparison tables? You used Llama2 for planning based methods and you used Qwen2.5 for online RL methods.
> > > Are there specific reasons that you use Llama2 instead of Llama3, Qwen 2.5 instead of Qwen3?
> > >
> > >
> > > Thank you for the follow-up question.
> > >
> > > Due to the limited time during the rebuttal period, we mainly followed prior work when choosing the backbone models, so that we could reuse established implementations and reported results for a more reliable comparison.
> > >
> > > Specifically, for the planning-based methods, we followed the setup used in MPO [1] and borrowed the corresponding baseline results, also in line with Reviewer 9jDb’s suggestion. For the online RL methods, we followed [GiGPO’s official implementation](https://github.com/langfengQ/verl-agent) [2] and used Qwen2.5-7B-Instruct, while re-running the experiments under our evaluation setting.
> > >
> > > In addition, Qwen2.5 has been open-sourced for a longer time, has stronger ecosystem and library support, and is itself a widely adopted baseline.
> > >
> > > Thank you for your interest and continued effort in helping us improve our work. If you have further questions, please let us know. We are happy to address any concern thoroughly and better demonstrate the merits of our paper during the rebuttal.
> > >
> > >
> > > [1] Boosting LLM Agents with Meta Plan Optimization. Weimin Xiong et al. Findings of EMNLP 2025.
> > > [2] Group-in-group policy optimization for llm agent training. Lang Feng et al., NeurIPS 2025.

---

### Official Review · Reviewer_f2M8 · 2026-03-11

**Soundness:** 3
**Presentation:** 3
**Significance:** 2
**Originality:** 2
**Overall Recommendation:** 4
**Confidence:** 4

**Summary:**

The paper proposes Q-Evolve, a self-evolving framework for training LLM agents under sparse episodic rewards. The method starts from behavior cloning on expert demonstrations, builds a hybrid offline dataset by mixing expert and self-collected trajectories, learns an in-distribution critic with a weighted IQL objective, derives step-wise process rewards via GAE, and updates the policy using a behavior-proximal objective with asymmetric clipping. The policy, critic, and offline dataset are then iteratively refreshed to enable self-evolution.

**Compliance With Llm Reviewing Policy:**

Affirmed.

**Final Justification:**

My main initial concerns were about the wording of the “in-distribution” claim, the justification for several heuristic design choices, and the strength of the sample-efficiency comparison. The rebuttal addressed these points well. Overall, the rebuttal improved my assessment, and I now view the paper as a solid contribution with practical value despite some remaining empirically motivated design choices.

**Key Questions For Authors:**

1. Can the authors clarify what exactly "in-distribution" or "within data support" means in the setting of natural-language action generation, and whether they have diagnostics showing that Q-Evolve reduces out-of-distribution action or state drift relative to the baselines?

2. How sensitive are the gains to the hand-designed reward penalties and to the weighted-IQL reweighting rule? Additional sensitivity analysis on these heuristics would strengthen the paper.

3. Can the authors provide a clearer compute comparison across methods, including total environment interactions, number of self-evolution loops, and training cost, especially for the sample-efficiency claims against QLASS and PPO?

**Limitations:**

The paper includes an impact statement, but it is mostly positive. I would encourage the authors to discuss failure modes more directly, especially risks from autonomous web or embodied agents, brittleness of heuristic reward shaping, and possible instability as self-evolution continues for more iterations.

**Strengths And Weaknesses:**

Strenths:
- Framing the problem of learning reliable long-horizon LLM under sparse delayed rewards as a closed loop of offline critic learning + process-reward labeling + in-distribution policy improvement is well motivated and conceptually coherent.
- The paper provides useful implementation details, including dataset statistics, hyperparameters, and a brief runtime analysis.

Weaknesses:
- Some of the paper’s strongest wording around being “strictly within offline data support” feels overstated to me. The inner loop is data-constrained, but the full method still collects fresh trajectories between iterations, and the notion of “support” is less clear when actions are free-form language/token sequences rather than a small discrete action set.
- Several design choices are somewhat heuristic, including the retrospective reward penalties, the weighted-IQL reweighting formula, the exclusion of auxiliary rewards from GAE despite using them for critic training, and the asymmetric clipping rule. These choices appear empirically useful, but the justification is mostly empirical.

---

> ### Author Rebuttal · Authors · 2026-03-31
>
> We thank the reviewer for the time and thoughtful suggestions. We address the concerns point by point.
>
> # Overstated "strictly within offline data support", Clarification of "in-distribution" and "within data support"
>
> Thanks for your comments. We explain as follows.
> **Explain of In-distribution & Data Support:** we would like to first to clarify that, both terms are defined with respect to the optimization procedure, rather than claiming that the learned policy is prevented from generating OOD natural-language actions at test time.
>
> More specifically,  both two terms mean that, for each loop, the critic is trained on observed state–action pairs in the fixed pre-collected dataset, and the policy is also optimized on those same labeled trajectories, via two ways.
> - First, during critic learning, we adopt IQL-style estimation, which avoids generating new candidate actions.  In vanilla Q-learning, the target involves a maximization over newly generated actions, given next state. In contrast,  IQL-style critic learning relies on expectile regression to approximate the explicit maximization, enabling critic learning using only the observed states and actions.
> - Second, during policy learning, process rewards are assigned to the observed trajectories and the policy is optimized directly on these labeled state–action pairs, rather than calculating process rewards on newly explored actions from online rollout.
>
>
> **Overstated "strictly within offline data support"** Thanks for pointing this out. We will revise it into "in-distribution optimization" to better clarify that our motivation is to perform process reward assignment and policy optimization over seen data distribution in each evolving iteration.
>
> ---
>
> # Heuristic design & Empirical justification
> We would like to clarify that these designs are not arbitrary, but are guided by the following considerations and then are validated by the empirical results:
> * Retrospective rewards can provide additional signals for invalid or non-executable actions, complementing the trajectory-level delayed rewards. We show that: (1) within a single loop, this design improves performance on ALFWorld (unseen) from 82.7 to 86.6 in one policy improvement; (2) across loops, the agent's format error rate is reduced from 16.87% to 11.82%.
> * Weighted IQL reweighting prioritizes transitions from trajectories with non-zero rewards, avoiding the propagation of zero rewards along trajectories and improving credit assignment under sparse rewards. This improves performance on ALFWorld (unseen) from 76.1 to 86.6.
> * Excluding auxiliary rewards from GAE avoids biasing policy updates toward heuristic signals, while still leveraging them to stabilize critic training. This raises performance on ALFWorld from 82.8 to 86.6.
> * Asymmetric clipping aims to address overfitting to expert trajectories. We also conduct ablations on different clipping values in Appendix Table 7.
>
> We will revise the paper to more clearly articulate these motivations.
>
> ---
>
> # Sensitivity on reward penalties and weighted-IQL
> Thanks for this insightful comment. We conduct sensitivity analysis as following.
>
> **Reward penalties.**  We test another reward setting (“RR-alt”) of different retrospective penalty values:  $r^{fmt}=-0.1$ for format error, $r^{inv}=-0.05$ for invalid action and $r^{rep}=-0.03$ for repeated action.
> For **weighted IQL**, in addition to removing the weighting mechanism entirely (“w/o W-IQL”), we further analyze two reduced variants: “W-IQL w/o temp” and “W-IQL w/o succ”, which removes the temporal re-weighting term and success-based re-weighting term separately.
> |  | Seen | Unseen |
> |-|-|-|
> | w/o RR | 83.6 | 82.7 |
> | RR-alt | 85.7 | 84.3 |
> | w/o W-IQL | 83.6 | 76.1 |
> | W-IQL w/o temp | 83.6 | 79.1 |
> | W-IQL w/o succ | 85.0 | 83.6 |
> | Ours | 87.9 | 86.6 |
> These results on ALFWorld show that our method is relatively robust to the design of reward penalties and both temporal re-weighting and success-based re-weighting contribute to the final better process reward estimation.
>
> ---
>
> # Sample-efficiency claims against QLASS and PPO
> Thanks for your question. We would like to refer the reviewer to our response to Reviewer 37HU: *Missing discussion of sample complexity & data efficiency* and *Scalability & Memory/time Cost*.
>
> ---
>
> # More detailed limitation:
> Thanks for pointing this out. We will include these risks more explicitly:
> - For autonomous web or embodied agents, while we propose simple and generally applicable rules (e.g., invalid actions, execution errors, repeated states), the examination method (like the structure of available feedback signals) may vary across tasks.
> - Second, evolution may not improve indefinitely and also may suffer from sub-optimal due to narrow exploration with the greedy policy. Therefore, designing exploration policies is also a good way to further improve the agent performance.
>
> ---
>
> We thank the reviewer again and look forward to further discussion.

---

> > ### Author Rebuttal · Reviewer_f2M8 · 2026-04-04
> >
> > Thank you for the detailed rebuttal. The response helped clarify several of my concerns, but I do not think they are fully resolved yet.
> >
> > Q1. I appreciate the clarification that “in-distribution” and “within data support” are defined with respect to the optimization procedure, rather than as a guarantee that the learned policy will not generate OOD natural-language actions at test time. This makes the intended claim much clearer, and I agree that the wording in the paper should be revised accordingly. However, I still think the paper should be careful not to overstate what this notion of “support” means in the setting of free-form language actions.
> >
> > Q2. The additional explanation and sensitivity analysis on the heuristic components are useful, especially for the retrospective reward penalties and weighted-IQL design. These results improve my confidence that these choices are not arbitrary. That said, the overall justification still seems primarily empirical, and some design choices remain only partially analyzed.
> >
> > Q3. The added table is helpful. Still, I do not think the efficiency comparison is fully nailed down yet. Some of the reported numbers appear to be summary-level estimates rather than fully standardized apples-to-apples measurements under a common protocol. PPO/RLOO/GRPO are given as approximate values, and their environment-step counts are derived from a generic rollout formula rather than presented as directly matched measurements in the same setup. And the numbers for the proposed method are reported as formulas depending on the number of loops rather than a single concrete setting aligned with the baselines. Since the paper makes explicit sample-efficiency claims against QLASS and PPO-style methods, I think this part should be made more precise in the final version.

---

> > > ### Author Response · Authors · 2026-04-05
> > >
> > > Thank you for the helpful follow-up question. We are happy to further clarify these points.
> > >
> > > ---
> > > # Q1
> > > Thanks for your instructive suggestion. We agree that, due to the large compositional vocabulary-level action space, this point requires careful wording. As noted, we will use “in-distribution optimization” precisely and avoid confusion by 'offline data support' and 'support'.
> > >
> > > ---
> > > # Q2
> > > Thanks for this comment.  Except from the empirical evidence, we provide a more detailed analysis of the motivation of components.
> > >
> > > **Retrospective rewards** provide additional signals for invalid or non-executable actions, complementing delayed trajectory-level rewards. We suggest that auxiliary rewards should be:
> > > * small relative to the extrinsic reward, non-dominating in cumulative return, and used mainly for shaping rather than overriding the task objective;
> > > * ranked by failure severity: format errors are easiest to avoid (highest penalty); non-executable actions require basic environmental understanding; repeated actions require more complex reasoning.
> > >
> > > **Weighted IQL reweighting** is used to upweight
> > > * later transitions: they are closer to terminal outcomes and have a shorter remaining bootstrap horizon, so their temporal-difference targets are less affected by long-range propagation noise;
> > > * successful trajectories: under sparse episodic rewards, they are a small but disproportionately informative subset, since most failure trajectories provide little direct reward information, while successful ones are precisely those that reach rewarding terminal states.
> > >
> > > Hence, reweighting increases the effective mass of reward-carrying transitions and prevents critic learning from being dominated by abundant low-signal failures.
> > >
> > > **Excluding auxiliary rewards from GAE avoids changing the policy objective**, and below is the theoretical analysis:
> > > Let $\delta_t^{env}=r_{t+1}^{env}+\gamma V(s_{t+1})-V(s_t)$, $A_t^{env}=\sum_{l=0}^{T-t-1}(\gamma\lambda)^l\delta_{t+l}^{env}$.
> > > If auxiliary reward is also included, then $\delta_t^{full}=\bigl(r_{t+1}^{env}+r_{t+1}^{aux}\bigr)+\gamma V(s_{t+1})-V(s_t)$, which gives $A_t^{full}=A_t^{env}+\sum_{l=0}^{T-t-1}(\gamma\lambda)^lr_{t+1+l}^{aux}$.
> > > Thus, including $r_{\mathrm{aux}}$ directly adds a discounted auxiliary-return term to the policy-side advantage target.
> > > Note $\lambda$ is the standard GAE decay parameter controlling the bias--variance tradeoff. When $\lambda=1$, GAE reduces to the Monte-Carlo advantage, and $A_t^{env}=\sum_{l\ge0}\gamma^lr_{t+1+l}^{env}-V(s_t)$, where the $V$-terms telescope and $V(s_T)=0$. Therefore, using only $r_{env}$ keeps the policy update aligned with the original environment-return objective and does not change the optimal policy.
> > > For $\lambda<1$, GAE no longer exactly equals the Monte-Carlo return advantage, but it still defines an environment-reward $\lambda$-return surrogate. The key distinction above remains exact: $A_t^{full}-A_t^{env}=\sum_{l=0}^{T-t-1}(\gamma\lambda)^lr_{t+1+l}^{aux}$. Hence, excluding it keeps policy learning aligned with the environment-reward surrogate alone.
> > >
> > > **Asymmetric clipping** reduces overfitting in offline policy learning by making *probability increases more conservative* while allowing *stronger suppression of negative-advantage actions*. In our BPPO-style objective, a smaller upper clip $\epsilon_{high}$ limits how aggressively the policy can increase probabilities of dataset actions, while a larger lower clip $\epsilon_{low}$ enables stronger down-weighting of actions with negative process rewards. This matters because offline overfitting often appears as **blind imitation of dataset actions**, including expert or frequent actions that may still be suboptimal under the learned process-reward signals. By tightening upward updates and allowing stronger downward corrections, asymmetric clipping reduces imitation bias and improves generalization.
> > >
> > > This interpretation is consistent with the ablation in Appendix Table 7: larger $\epsilon_{low}$ is more beneficial than larger $\epsilon_{high}$, and the best performance is achieved at $\epsilon_{low}=0.8,\epsilon_{high}=0.4$, which yields 87.9 / 86.6 on ALFWorld seen/unseen.
> > >
> > > ---
> > > # Q3
> > > To compare more directly, we summarize a more direct comparison in AlfWorld:
> > > | | Init | Env Steps |Seen| Unseen |
> > > |-|-|-|-|-|
> > > |PPO|-|320K|59.4|67.7|
> > > |RLOO|-|320K|56.4|36.6|
> > > |GRPO|-|320K|39.7|32.2|
> > > |SFT|Expert|-|74.9|62.3|
> > > |SFT+PPO|Expert|320K|72.6|77.6|
> > > |SFT+RLOO|Expert|320K|75.0|51.4|
> > > |SFT+GRPO|Expert|320K|66.7|74.1|
> > > |Ours (1-iter)|Expert|13K|88.6|87.3|
> > >
> > > where “Env Steps” denotes the number of samples used. Thus, the table directly supports our sample-efficiency claim: online methods use 320K environment steps, while our method (1-iter) uses only 13K and achieves the best performance.
> > >
> > > ---
> > >
> > > Thank you for raising these question. We will revise the paper to include the above clarification.  We hope this addresses your concern, and we would be happy to discuss this further.

---

### Official Review · Reviewer_37HU · 2026-03-12

**Soundness:** 3
**Presentation:** 2
**Significance:** 3
**Originality:** 4
**Overall Recommendation:** 5
**Confidence:** 3

**Summary:**

This paper introduces Q-evolve, a hybrid offline/online learning algorithm for LLM web agents. The paper presents a few stages of the algorithm in order to achieve the full performance, including reward labeling, IQL training, and behavior cloning from offline data. The paper then demonstrates good results on three tasks, with greater sample efficiency.

**Compliance With Llm Reviewing Policy:**

Affirmed.

**Final Justification:**

The authors satisfied my concerns during the rebuttal phase and I wish to keep my positive score.

**Key Questions For Authors:**

- Can you discuss more regarding memory increase and time increase?
- After iteration 2, is the model unable to continue evolving? Does this not go counter to the self-evolving loop?
- Is this algorithm more data efficient than other models?

**Limitations:**

There is no discussion of overhead and potential extra memory of training 2 q functions at the same time, which is non-trivial in many applications.

**Strengths And Weaknesses:**

**Strengths**
- This algorithm seems sufficiently novel, and introduces a few nice tricks (weighted iql, using GAE with only environment rewards, etc)
- This paper nicely ablates many of the design choices.
- The paper is relatively well written.

**Weaknesses**
- The paper is lacking a section which introduces the flow of the algorithm. For a relatively complex algorithm flow, Figure 2 is the only figure that fully goes through the pattern. Section 4 before 4.1 attempts to do this, but explanation could be clearer.
- It would be nice to have discussion about sample complexity. Although the algorithm seems to convincingly outperform other methods, this is alluded to, but never fully discussed.
- This method requires a lot of overhead in multiple Q functions and a policy. I worry about scalability of this method, and in particular memory/time cost. It would be nice to discuss this.

---

> ### Author Rebuttal · Authors · 2026-03-31
>
> We sincerely thank the reviewer for the positive support. We address the questions and concerns point by point.
>
> ---
> # Missing Algorithm Overview
> > Lacking a section for introducing the algorithm flow.
> Thanks for your suggestion. We agree that the overall algorithmic flow can be presented more clearly.
>
> To address this, we will revise Section 4 to include a more structured and high-level overview of the full pipeline before introducing the individual components. Specifically, we will explicitly organize the method as an interactive loop, where each iteration consists of four stages: (1) hybrid data construction (expert + agent trajectories), (2) in-distribution critic learning via weighted IQL, (3) process reward derivation through advantage estimation (GAE), and (4) in-distribution policy optimization via BPPO. We will further clarify how these stages are iteratively applied to refine the agent and include concise pseudo-code in the appendix to make the execution order and dependencies fully explicit.
>
> We will also revise Figure 2 to highlight the single loop illustration and interactive optimization, as well as make the caption more self-contained (please see [revised Fig. 2](https://bashify.io/i/gTF49G)).
>
> ---
> # Missing discussion of sample complexity & data efficiency
> Thanks for your suggestion. We agree that a clearer discussion of sample complexity is important.
>
> We would like to clarify that our method is substantially more sample-efficient than existing approaches. As shown in Figure 1, our method achieves a score of 90 with only 20K environment steps, whereas QLASS achieves a score of 80 with 600K steps.
>
> Additionally, QLASS incurs extra inference-time cost by evaluating multiple candidate actions (e.g., ≥2 per step) with the learned critic.
>
> We will also include the comparison with the online RL methods (e.g., PPO/GRPO/RLOO), which typically require around 960K environmental steps (150 rollout steps × 128 trajectories × 50 timesteps in [Verl-agent](https://github.com/langfengQ/verl-agent)). We summarize all the comparison in the following table.
> | Method | Runtime | Environment Steps | Memory Cost | Inference |
> |-|-|-|-|-|
> |PPO| ~32h|960K|x3|None|
> | RLOO|~30h|960K|x2|None|
> | GRPO| ~28h|960K|x2|None |
> | QLASS| -|600K|x1| multi-candidate calculation and scoring |
> | Ours | 13h + 3(N-1) | 13K + 7K * (N - 1)|x2| None |
> where N is the number of interactions of our method, in practice we choose N=2 or N=3. 'x3' denotes loading 3 LLMs in parallel during training.
>
> Additionally, we compare the performance of those online RL baselines. Please refer to the response to Reviewer ZCHC: *Results with various model architectures and model sizes*.
>
> These comparisons highlight that Q-Evolve achieves competitive or superior performance while using significantly fewer environment interactions and avoiding expensive online exploration or search-based evaluation.
>
> We will include the above in our revision.
>
> ---
>
> # Scalability & Memory/time Cost (multiple Q functions)
> Multiple Q function does not introduce heavy computation burden because:
> - During critic learning, the value function uses one LLM backbone, while the Q-function and target Q-function share another LLM backbone; thus, critic training is effectively implemented with two critic models rather than three fully separate models.
> - During policy learning, the advantages are pre-computed offline, so no critic needs to be kept in memory. At that stage, training only involves the policy model and the reference model.
>
> Therefore, the memory footprint of our policy optimization stage is similar to GRPO, with only two models loaded at the same time. We also compare the memory cost and time consumption in the response to *Missing discussion of sample complexity & data efficiency*.
>
> As a consequence, our method is scalable and achieves lower overall cost by (i) avoiding large-scale online interaction, (ii) eliminating the need for inference-time reward model evaluation, and (iii) sharing parameters across critic components.
>
> ---
>
> # Evolution Loops Stopping
> > After iteration 2, is the model unable to continue evolving? Does this not go counter to the self-evolving loop?
> Thanks for your question. In principle, the interactively improving is not assumed to be indefinite, it has a practical performance ceiling, like ETO.
>
> In our experiments, after certain iterations, we found no obvious improvement of agent performance on training task set. Considering the additional cost of training the critic and the policy in another loop, we therefore stopped the evolution at that point.
>
> Figure 3 only shows the first two iterations and we use 3 interactions for Webshop and 2 for others. We will clarify in the revision that the number of self-evolving loops is a tunable choice determined by the empirical performance–cost trade-off, rather than a fixed property of the method.
>
> ---
> We thank the reviewer again and we look forward to further discussion.

---

> > ### Author Rebuttal · Reviewer_37HU · 2026-04-03
> >
> > I thank the authors for their rebuttal.
> >
> > After reading the other reviewers comments and the rebuttal, I will maintain my positive score as my comments are addressed. However, I think two parts of this paper can be improved:
> > 1. clarify of the algorithm and manuscript
> > 2. explanation of heuristics (eg, reward values)
> >
> > However, I expect these to be rewritten before a camera-ready submission.

---

> > > ### Author Response · Authors · 2026-04-04
> > >
> > > We are very encouraged with your possitive support!
> > >
> > > We agree with your suggestion to make these two improvement in our paper. Therefore, we plan to add:
> > > - more clear overall pipeline illustration, including the [revised Fig. 2](https://bashify.io/i/gTF49G), better clarification of input/output for each component and the pseudocode in our response to the follow-up Q2 of Reviewer 9jDb.
> > > - more explanation of the heuristics
> > >     - the reward values, as we discussed in the response to Reviewer 9jDb (Choice of restropective rewards & sensitivity, generality, follow-up Q1), Reviewer f2M8 (Heuristic design & Empirical justification), Reviewer ZCHZ (Specific numerics in the retrospective reward)
> > >     - the core motivations and the motivation of all the components, as discussed in the response to Reviewer 9jDb (A combination based on "trial and error", follow-up Q3), Reviewer f2M8 (Heuristic design & Empirical justification)
> > >
> > > Thanks again for your effort in reviewing our paper and the constructive feedback.

---

### Decision · Program_Chairs · 2026-04-30

**Decision:**

Accept (regular)

**Comment:**

The reviewers initially had concerns on the complexity of the proposed methods and the lack of experiments (such as ablations and model architectures). Most of these concerns were addressed during the rebuttal and most reviewers became positive about the paper. There's one negative reviewer left. The remaining concerns are reward design ablation, the complexity of algorithm design (a A+B+C+D method), and the paper title is over claiming. In the discussion phase, the authors provided additional ablation on reward design, and promised to revise the paper title to be ``self-evolved LLM agents with in-distribution process rewards''. I agree with the negative reviewer that the wording of ``offline'' can be misleading, given that this regime here is more similar to off-policy RL rather than offline RL due to the iterative collection of new samples. Please revise the paper accordingly to be more accurate about the scope and contribution. As for the second point, my take is that a complex method is acceptable if that is well motivated and systematically tested, and this is what most reviewers agree. Therefore, I would recommend acceptance of this paper.